# Inter-species association mapping links splice site evolution to METTL16 and SNRNP27K

**Matthew T Parker[1], Sebastian M Fica[2], Geoffrey J Barton[1], Gordon G Simpson[1,3]***

[1]School of Life Sciences, University of Dundee, Dundee, United Kingdom; [2]Department of Biochemistry, University of Oxford, Oxford, United Kingdom; [3]Cell & Molecular Sciences, James Hutton Institute, Invergowrie, United Kingdom

**Abstract** Eukaryotic genes are interrupted by introns that are removed from transcribed RNAs by splicing. Patterns of splicing complexity differ between species, but it is unclear how these differences arise. We used inter-species association mapping with Saccharomycotina species to correlate splicing signal phenotypes with the presence or absence of splicing factors. Here, we show that variation in 5′ splice site sequence preferences correlate with the presence of the U6 snRNA N6-methyladenosine methyltransferase METTL16 and the splicing factor SNRNP27K. The greatest variation in 5′ splice site sequence occurred at the +4 position and involved a preference switch between adenosine and uridine. Loss of METTL16 and SNRNP27K orthologs, or a single SNRNP27K methionine residue, was associated with a preference for +4 U. These findings are consistent with splicing analyses of mutants defective in either METTL16 or SNRNP27K orthologs and models derived from spliceosome structures, demonstrating that inter-species association mapping is a powerful orthogonal approach to molecular studies. We identified variation between species in the occurrence of two major classes of 5′ splice sites, defined by distinct interaction potentials with U5 and U6 snRNAs, that correlates with intron number. We conclude that variation in concerted processes of 5′ splice site selection by U6 snRNA is associated with evolutionary changes in splicing signal phenotypes.

*For correspondence:
g.g.simpson@dundee.ac.uk

**Competing interest:** The authors declare that no competing interests exist.

## Editor's evaluation

The manuscript addresses the ways in which different organisms have evolved pre-messenger RNA systems that are either more or less complex, a question that underlies the evolution of complex organisms and the genome adaptation of simple organisms to their specific environments. This important manuscript provides the underlying molecular mechanisms of how 5' splice site sequence preference may have evolved, with solid structural modeling data in support.

## Introduction

Introns in genes are a defining characteristic of eukaryotes (*Madhani, 2013*). The excision of introns and the splicing together of flanking exon sequences from RNA Polymerase II-transcribed RNAs is mediated by dynamic assemblies of proteins and small nuclear RNAs (UsnRNAs) called spliceosomes (*Wilkinson et al., 2020*). The number of introns and the complexity of splicing patterns vary widely between species (*Irimia et al., 2007*; *Lim et al., 2021*; *Sales-Lee et al., 2021*). For example, human gene expression is characterised by a relatively large number of introns and spliceosomal proteins, degenerate splicing cis-elements, and complex patterns of alternative splicing choices on transcripts from the same gene (*Lee and Rio, 2015*; *Nilsen and Graveley, 2010*; *Sales-Lee et al., 2021*). In

contrast, most genes in the experimental model *Saccharomyces cerevisiae* lack introns; splicing cis-elements are almost invariant and alternative splicing is rare (*Neuvéglise et al., 2011*; *Plaschka et al., 2019*; *Sales-Lee et al., 2021*). Alternative splicing correlates with developmental complexity (*Bénitière et al., 2023*; *Bush et al., 2017*; *Chen et al., 2014*; *Wright et al., 2022*). Yet some developmentally simple fungal species, including *Cryptococcus neoformans,* have many introns, degenerate splicing signals, and a near complete repertoire of human spliceosomal protein orthologs (*Csuros et al., 2011*; *Lim et al., 2021*; *Mitrovich et al., 2010*; *Sales-Lee et al., 2021*). The simplification of splicing detected in *S. cerevisiae* must therefore have occurred through the evolutionary loss of some introns and splicing factors (*Csuros et al., 2011*; *Irimia et al., 2007*; *Lim et al., 2021*; *Mitrovich et al., 2010*; *Rogozin et al., 2012*; *Schwartz et al., 2008*). It is unclear what drives global splicing patterns to become either more or less complex or how these changes in complexity are achieved (*Bénitière et al., 2023*; *Bush et al., 2017*; *Wright et al., 2022*).

Splicing proceeds through two sequential transesterification reactions (*Wilkinson et al., 2020*). The first reaction, branching, occurs via nucleophilic attack of the 5' splice site (5'SS) by the 2' hydroxyl of the intron branch point adenosine. The second reaction, exon ligation, occurs via nucleophilic attack of the 3' hydroxyl of the 5' exon on the 3' splice site (3'SS). Spliceosomes undergo sequential conformational and compositional changes as they process introns. During initial assembly, potential 5'SSs are typically recognised by U1 snRNA, and the branch point adenosine and 3'SS are recognised cooperatively by binding of the SF1 and U2AF proteins, and U2 snRNA (*Fica, 2020*). 5'SSs are subsequently transferred to U5 and U6 snRNA. In humans, 5'SS transfer is decoupled from active site formation, potentially enabling plasticity in 5'SS selection (*Charenton et al., 2019*; *Fica, 2020*). U5 snRNA loop 1 binds to the exon upstream of the 5'SS. The conserved ACAGA sequence of U6 snRNA pairs with the intron sequence adjacent to the 5'SS. The central adenosine of human U6 snRNA AC**A**GA is methylated at the N6 position (m6A) by METTL16 (*Aoyama et al., 2020*; *Pendleton et al., 2017*; *Warda et al., 2017*). Cryogenic-electron microscopy (cryo-EM) structures show that human U6 snRNA m6A$_{43}$ faces the +4 position of the 5'SS (which is generally A in humans) in a trans Hoogsteen sugar edge interaction (*Bertram et al., 2017*; *Parker et al., 2022*). m6A-modification of the U6 snRNA AC**A**GA sequence is found in diverse eukaryotes. For example, the corresponding U6 snRNA nucleotide is m6A-modified in *Caenorhabditis elegans*, *Schizosaccharomyces pombe* and *Arabidopsis thaliana* by METTL16 orthologs (*Ishigami et al., 2021*; *Mendel et al., 2021*; *Wang et al., 2022*). In contrast, cryo-EM structures of *S. cerevisiae* spliceosomal complexes reveal major differences with humans in these early stages of splicing (*Charenton et al., 2019*). In *S. cerevisiae*, the transfer of 5' SSs from U1 to U5 and U6 snRNA is directly coupled to active site formation. U5 snRNA loop 1 pairs with degenerate upstream exon sequences while the ACAGA box of U6 snRNA pairs with almost invariant intron sequences. There is no METTL16 ortholog in *S. cerevisiae* and the ACAGA sequence is not m6A-modified (*Morais et al., 2021*). The central adenosine of *S. cerevisiae* U6 snRNA ACAGA does not face a 5'SS +4 A but instead makes a Watson-Crick base pair with the almost invariant +4 U of *S. cerevisiae* 5'SSs (*Neuvéglise et al., 2011*; *Wan et al., 2019*). Therefore, these early steps in 5'SS selection exemplify distinctions between species that differ not only in terms of developmental complexity but also patterns of splicing complexity.

The direct impact of U6 snRNA m6A modification on splicing in humans remains uncharacterised. However, recent analyses of mutants lacking METTL16 orthologs in *S. pombe* (*mtl16Δ*) and *Arabidopsis* (*fio1*) have revealed that U6 snRNA m6A modification has profound effects on the accuracy and efficiency of splicing (*Ishigami et al., 2021*; *Parker et al., 2022*). Global splicing analyses of *S. pombe mtl16Δ* and *Arabidopsis fio1* mutants show that 5'SSs with +4 A are most sensitive to the loss of U6 snRNA m6A modification. In contrast, splicing became more efficient at 5'SSs with +4 U in *S. pombe mtl16Δ* and *Arabidopsis fio1* mutants. Relatively strong interactions with U5 snRNA can compensate for weakened U6 snRNA interactions caused by the loss of U6 snRNA m6A modification (*Ishigami et al., 2021*; *Parker et al., 2022*). A negative correlation between U5 and U6 snRNA interaction potentials reveals that there are two major classes of 5'SSs (*Parker et al., 2022*). Since in some species, pairs of alternative 5'SSs tend to be of opposite classes, this genomic feature may influence alternative splicing.

Given the similarity of the 5'SS +4 consensus preferred in *Arabidopsis fio1* mutants to that observed in *S. cerevisiae,* we speculated that the loss of a METTL16 ortholog in some species could contribute to an evolutionary change in 5'SS sequence preference. To address this possibility, we

took a phylogenetics approach by examining how 5'SS sequence preference changed in different species. We applied inter-species association mapping as an unbiased approach to identify factors correlated with 5'SS sequence (*Kiefer et al., 2019*; *Smith et al., 2020*). This approach is conceptually related to a genome-wide association study (GWAS) in which genomic variation is correlated with phenotype. However, rather than consider variation within a species, inter-species association mapping correlates a phenotype with genotypic variation between species, usually at the level of gene presence or absence, whilst correcting for the relatedness of species (*Kiefer et al., 2019*; *Smith et al., 2020*). We surveyed the genomes of more than 200 species from the Saccharomycotina clade of fungi and found multiple independent switches in sequence preference at the 5'SS +4 position. We found that the identity of the nucleotide at the 5'SS +4 position is most strongly correlated with the presence or absence of METTL16. In addition, we detected a conserved methionine in the spliceosomal protein SNRNP27K that also correlates with 5'SS +4 sequence preference. We conclude that variation in factors that act in concert to influence U6 snRNA interactions with 5'SSs are associated with evolutionary change in splicing signal phenotypes.

## Results

### METTL16 is widely conserved in eukaryotes

Patterns of presence/absence variation of human spliceosomal protein orthologs in some fungal species have recently been reported (*Sales-Lee et al., 2021*). However, because m⁶A-modified U6 snRNA is a component of spliceosomes but METTL16 is not, variation in METTL16 or UsnRNA modification was not assessed in this context. To understand the variation in METTL16 across eukaryotes, we used the software tool phmmer (*Eddy and Pearson, 2011*) to search Uniprot databases (*Bateman et al., 2023*) for proteins containing similarity to the METTL16 methyltransferase domain (MTD) in 33 diverse eukaryotic species from Metazoa, Fungi, Archaeplastida, Chromista, and Excavata. Putative METTL16 orthologs were identified in species from all groups except Excavata (*Figure 1A*). We were unable to detect METTL16 orthologs in four species, but since these species were distributed across the kingdoms, we conclude that METTL16 is ancestral to most or all eukaryotes and that absences are the result of independent gene loss events.

To consider potential changes in METTL16 substrate specificity, we analysed the domain structure of the METTL16 orthologs in more detail. Structural studies indicate that human METTL16 is comprised of two globular domains: the N-terminal MTD and a C-terminal domain called the 'vertebrate conserved region' (VCR). Both domains contribute to METTL16 U6 snRNA m⁶A-modification specificity (*Aoyama et al., 2020*). Since primary sequence alignments fail to identify the METTL16 VCR in more distantly related species (*Aoyama et al., 2020*; *Ju et al., 2023*; *Pendleton et al., 2017*), we developed a revised approach to this search by incorporating protein structure predictions. Human METTL16 VCR has structural and functional equivalence to the KA-1 domain of the U6 3' uridyltransferase TUT1 (*Aoyama et al., 2020*). The METTL16 VCR and TUT1 KA-1 domains bind the conserved internal stem-loop of U6 snRNA to enhance substrate specificity (*Aoyama et al., 2020*; *Yamashita et al., 2017*). Given the low sequence similarity between METTL16 VCR and TUT1 KA-1, we asked whether METTL16 orthologs contain topologically similar C-terminal domains that are poorly conserved at the sequence level. To address this question, we used Alphafold2 predictions (*Varadi et al., 2022*) of the 29 orthologous METTL16 proteins identified in the 33 species from across Eukaryota. We used the predicted local distance difference test (pLDDT) and predicted alignment error (PAE) values provided for each model to segment the proteins into globular domains, excluding regions predicted to be flexible or disordered (*Oeffner et al., 2022*). Using this method, all the METTL16 orthologs were found to have either one or two globular domains (*Figure 1A*). We then superimposed all pairs of domains and calculated the template modelling score (TM-score), which is a measure of the similarity of two or more protein folds (*Zhang and Skolnick, 2005*). As positive controls, we included structures of the human METTL16 MTD and VCR domains, and TUT1 KA-1 domain, determined by X-ray crystallography (*Aoyama et al., 2020*; *Ruszkowska et al., 2018*; *Yamashita et al., 2017*). All predicted METTL16 ortholog domains were classified as either MTD-like, VCR/KA-1-like, or MTD +VCR/KA-1-like (*Figure 1B*), with all MTD-like and VCR-like domains superimposing onto the experimentally determined human METTL16 MTD or VCR/KA-1 structures, respectively, with TM-scores that indicate structural analogy (*Figure 1C*). The MTD +VCR/KA-1-like

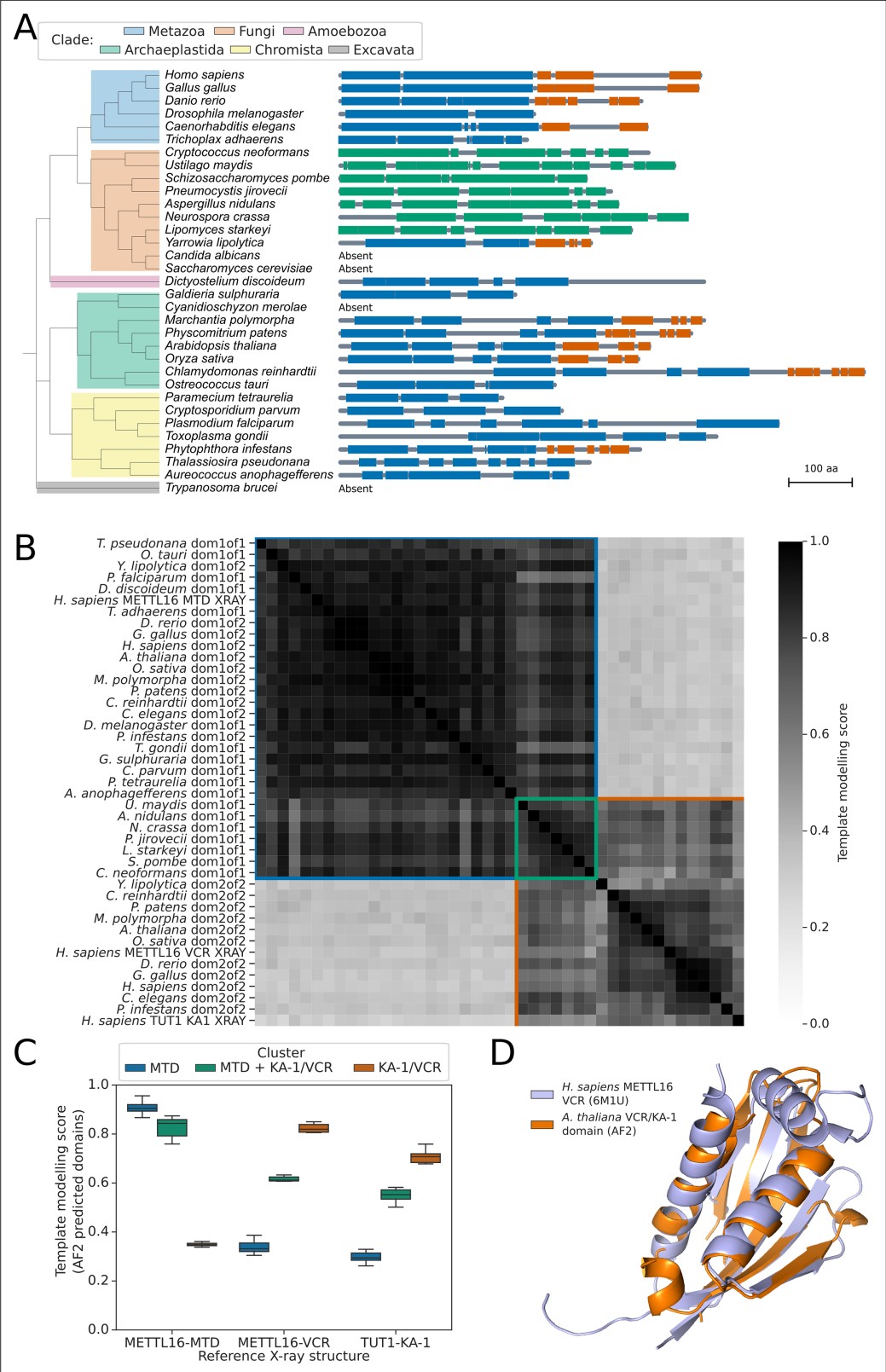

**Figure 1.** METTL16 is widely conserved across eukaryotes. (**A**) Phylogenetic tree showing the presence and absence of a METTL16 ortholog in 33 eukaryotic species. Linear protein structures with globular domains identified from Alphafold2 models are shown on the right of the tree. Domains are colored by cluster: MTD domains in blue, MTD +VCR/KA-1 domains in green, and VCR/KA-1 domains in orange. Likely loop/disordered regions with

*Figure 1 continued on next page*

*Figure 1 continued*

low confidence predictions are shown as grey lines. (**B**) Distance matrix heatmap showing the pairwise TMscore of segmented domains from the Alphafold2 predictions of 29 METTL16 orthologs. The X-ray structures of human METTL16 MTD and VCR, and TUT1 KA-1 are included as positive controls. Domains are grouped into clusters (diagonal boxes) using the same color scheme as in (**A**). (**C**) Boxplot showing TMscores of segmented domains from Alphafold2 predictions of 28 METTL16 orthologs superimposed onto experimentally determined structures of human METTL16 MTD and VCR, and TUT1 KA-1. (**D**) Superimposition of the VCR/KA-1 domain of *Arabidopsis* FIO1 predicted by Alphafold2 onto the X-ray structure of the human METTL16 VCR.

The online version of this article includes the following figure supplement(s) for figure 1:

**Figure supplement 1.** Structural analysis of fungal METTL16 orthologs.

class is specific to the fungal species examined here and is explained by the close packing of MTD and KA-1 domains which prevents segmentation by PAE. This results in 'superdomains' that have structural analogy to both the MTD and KA-1 domains of human METTL16 (*Figure 1C*). The VCR/KA-1-like domain was absent from METTL16 orthologs in some species but could be identified in orthologs from across the eukaryotic tree of life, including *C. elegans, S. pombe, Arabidopsis,* and *Phytophthora infestans* (*Figure 1A*). We used the software tool US-align (Universal Structure alignment) (*Zhang et al., 2022*) to perform multiple structure alignments of all 11 VCR/KA-1-like domains predicted by Alphafold2, resulting in an overall TM-score of 0.53, which indicates that all the domains are structurally analogous (*Zhang and Skolnick, 2005*). For example, the predicted C-terminal domain of *Arabidopsis* FIO1 could be superimposed onto the crystal structure of the human METTL16 VCR domain with a TM-score of 0.82, and an RMSD of 0.86 Ångstroms at 59 structurally equivalent alpha carbon atoms (*Figure 1D*). All five beta strands and the two larger alpha helices of the METTL16 VCR domain are predicted by Alphafold2 to be conserved in FIO1.

The MTD +VCR/KA-1-like cluster of 'superdomains' from fungal species, for which the MTD and VCR/KA-1-like subdomains could not be segmented, superimposed well onto each other with an overall TM-score of 0.83 for 7 orthologs (*Figure 1B*). This indicates that the predicted relative positions of the MTD and VCR/KA-1-like subdomains within these superdomains is consistent across orthologs. Arginine 258 of *S. pombe* MTL16, which is located between the MTD-like and VCR/KA-1-like subdomains, is predicted to form hydrogen bonds with backbone carbonyl oxygens and a serine sidechain that binds the two subdomains together, helping to form the superdomain (*Figure 1— figure supplement 1A*). In agreement with this, Arg258 is strongly conserved in the 7 orthologs which have an MTD +VCR/KA-1-like superdomain but absent from *Yarrowia lipolytica* METTL16 (*Figure 1— figure supplement 1B*), which is segmentable into two domains (*Figure 1A*). The position of the VCR/ KA-1-like subdomain relative to the known ACAGA binding site of the MTD in these orthologs may help to improve our understanding of how the VCR/KA-1-like domain contributes to the specificity of METTL16 for U6 snRNA.

In summary, by incorporating protein structure predictions we could revise the understanding of the conserved domain structure of METTL16, detecting evidence for the VCR/KA-1 domain in species where it had previously been reported to be absent (*Mendel et al., 2021*). These results agree with recent structural studies of the *C. elegans* METTL16 ortholog METT10 which confirmed the presence of a KA-1 domain (*Ju et al., 2023*). Overall, our analysis suggests that the last eukaryotic common ancestor may not only have had a METTL16 ortholog but one that contained a KA-1-like domain too. Consequently, METTL16 with a domain that enhances substrate specificity for U6 snRNA is a feature of early eukaryotes.

## Variation in the identity of the 5'SS +4 position evolved independently on multiple occasions

To understand evolutionary changes in splicing complexity, we focused on splicing signal phenotypes in the Saccharomycotina clade of ascomycetous yeasts. The Saccharomycotina clade was chosen for three reasons. First, simplification of spliceosome composition, intron content and cis-element splicing signals has already been established in these species (*Lim et al., 2021*; *Neuvéglise et al., 2011*; *Sales-Lee et al., 2021*). Second, the Saccharomycotina clade includes *S. cerevisiae*, which is an important model species for the study of splicing. Finally, the availability of many sequenced

Saccharomycotina genomes has the potential to provide statistical power for inter-species association mapping.

We devised a new bioinformatics pipeline to identify groups of orthologous splicing factor genes (orthogroups) and splicing sequence phenotypes from genomic sequences deposited in NCBI (*Figure 2—figure supplement 1*). We used the software tool Funannotate (*Palmer and Stajich, 2020*) to annotate the genomes of 227 publicly available Sacchromycotina genomes (*Supplementary file 1*; *Dujon et al., 2004*; *Riley et al., 2016*; *Shen et al., 2018*; *Shen et al., 2016*). In addition, we incorporated an outgroup comprised of 13 well-annotated eukaryote genomes, including the human genome and 3 reference Saccharomycotina genomes (*S. cerevisiae, Candida albicans,* and *Yarrowia lipolytica*) (*Dujon et al., 2004*; *Engel et al., 2022*; *Muzzey et al., 2013*; *Nurk et al., 2022*). The outgroup helps root the phylogenetic tree and annotate the orthogroups predictions. Protein sequences from these annotations were clustered into orthogroups using the software tool Orthofinder (*Emms and Kelly, 2019*), which also generates a species tree (*Emms and Kelly, 2018*). The initial de novo annotations of some species contained many incorrectly annotated introns. Consequently, we used multiple sequence alignments of proteins from each orthogroup to filter introns, keeping only those introns that were conserved in the same multiple alignment column and frame in at least two species (*Rogozin et al., 2003*). Overall, the pipeline yielded sequence information on introns and identified orthologous groupings of protein-coding genes.

We used the conserved annotated introns to estimate the 5′SS sequence preference of each species and mapped these onto the species tree whilst also estimating ancestral 5′SS sequence preferences (*Figure 2A*; *Paradis et al., 2004*; *Schwartz et al., 2008*). This analysis identified limited variation in the frequency of the preferred nucleotides at the −1,+3, and +5 positions of the 5′SS (*Figure 2B*, *Figure 2—figure supplement 2A–C*). In contrast, there have been multiple independent changes of sequence preference at the 5′SS +4 position in the Saccharomycetaceae, Debaryomycetaceae (CUG-Ser1) and Pichiaceae families (*Figure 2C*). These preference changes almost exclusively comprise switches between +4 A and +4 U since there was much less variation in the ratio of W (A or U) to S (G or C) nucleotides (*Figure 2B*, *Figure 2—figure supplement 2D*). For example, the Saccharomycetaceae (which includes *S. cerevisiae*) and their sister taxa Saccharomycodaceae all have a similarly strong 5′SS preference for +4 U, suggesting that an overall preference for +4 U is ancestral to this group. However, the estimated next closest family to the Saccharomycetaceae, the Phaffomycetaceae, have an overall preference for +4 A, indicating that the invariant 5′SS +4 U phenotype of Saccharomycetaceae may have evolved since the divergence of the two families. Although the confidence of the relative arrangement of Phaffomycetaceae, Saccharomycetaceae and CUG-Ser1 clades on the species tree is low (*Figure 2A*), the placement of Phaffomycetaceae matches previous approaches that used genome-scale data (*Shen et al., 2020*; *Shen et al., 2016*). Furthermore, the preference for 5′SS +4 A in families basal to the CUG-Ser1 clade, such as Cephaloascaceae (e.g. *Cephaloascus fragrans*), corroborates the finding that the switch to +4 U seen in many genera of the Debaryomycetaceae family of the CUG-Ser1 clade (e.g. in *C. albicans*) is independent of Saccharomycetaceae (*Figure 2C*, *Figure 2—figure supplement 3*). This indicates that the invariant 5′SS +4 U preferences seen in *S. cerevisiae* and *C. albicans* are the result of convergent evolution, which contrasts with a previous interpretation made when fewer genomes were available for comparison (*Schwartz et al., 2008*). Therefore, the improved resolution that the recent availability of more genome sequences makes possible reveals that the evolutionary history of splicing signal phenotypes in fungi is more complex than previously recognised.

We explored whether there were changes in sequence preferences at the 3′SSs of the species in our dataset and detected evidence of switches in 3′SS sequence preference at the −3 position, between C and U. The variation in this phenotype was lower than that observed at the 5'SS +4 position (*Figure 2—figure supplement 4A–B*). Overall, we conclude that switches from 5′SS +4 A to +4 U preference have occurred independently on multiple occasions in the Saccharomycotina. Consequently, this suggests that Saccharomycotina species present variation in the 5′SS +4 phenotype suitable for inter-species association studies.

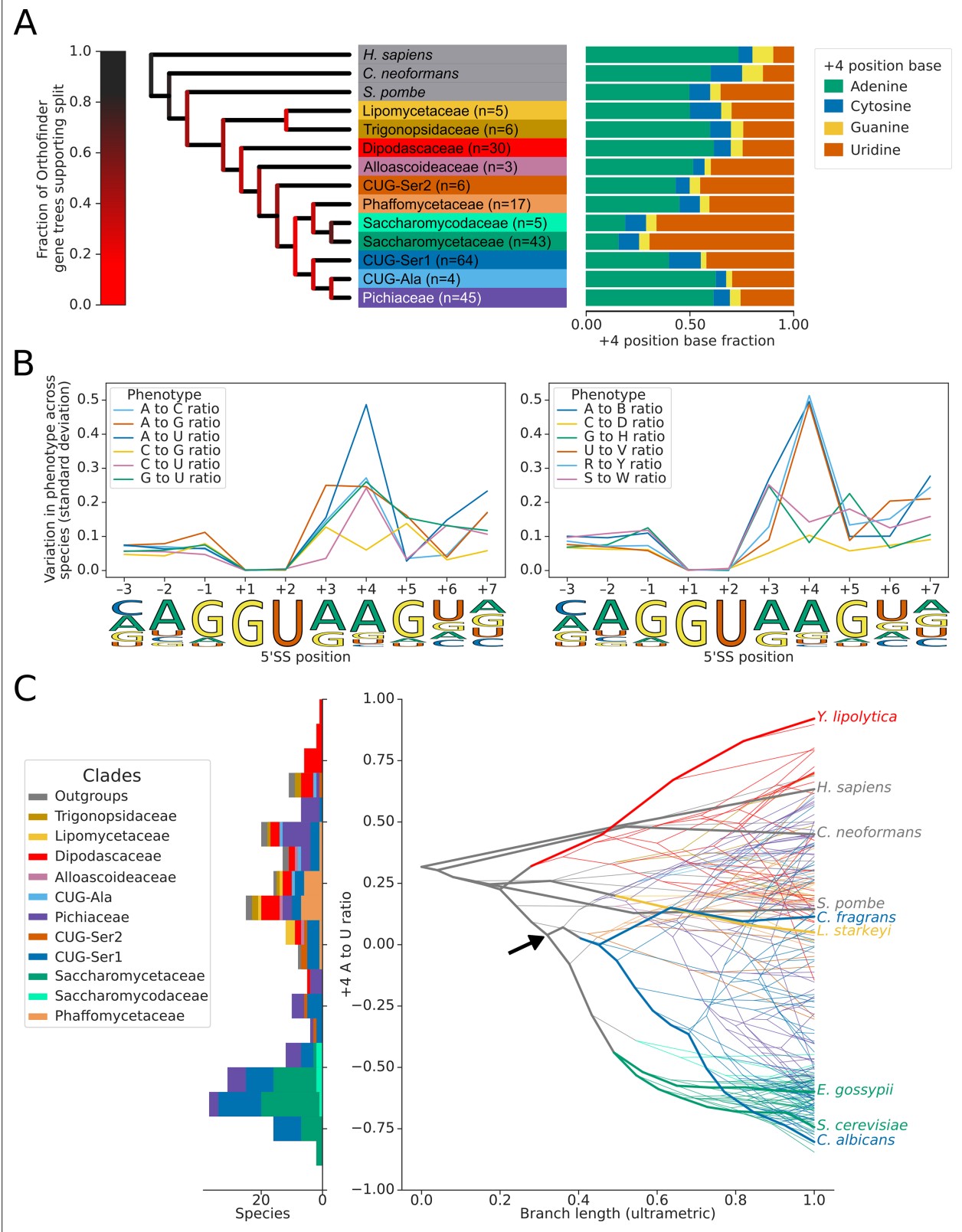

**Figure 2.** Variation in the identity of the 5' splice site +4 position evolved independently on multiple occasions. (**A**) Phylogenetic tree showing the estimated +4 position nucleotide fractions (as stacked bars) of the last common ancestors of 11 clades of Saccharomycotina, plus three outgroup species. The phylogenetic tree used is a collapsed ultrametric version of the species tree generated by Orthofinder from the proteomes of 240 species. Bifurcations are colored by confidence, calculated using the number of single-locus gene trees that support the bifurcation in the Orthofinder STAG

*Figure 2 continued on next page*

*Figure 2 continued*

algorithm (*Emms and Kelly, 2019*; *Emms and Kelly, 2018*) (this measure is generally more stringent than bootstrap values for trees generated using concatenated multiple sequence alignment and maximum likelihood methods). Ancestral nucleotide fractions were calculated using Sankoff Parsimony (*Schwartz et al., 2008*) (**B**) Line plots showing the standard deviation of nucleotide frequency ratio phenotypes for 240 species, across the –3 to +7 positions of the 5'SS. Left panel shows all pairwise combinations of single nucleotide frequency phenotypes (e.g. A to U ratio), right panel shows all single nucleotide versus other combinations (e.g. G to H [A, C or U]), as well as R (A or G) to Y (C or U) and S (G or C) to W (A or U) ratios. Human 5'SS consensus sequences are shown for each position as a guide. (**C**) Stacked histogram and traitgram showing the distribution of 5'SS +4 A to U ratio phenotypes in different Saccharomycotina species. The traitgram shows the predicted path of the 5'SS +4 A to U ratio phenotype through evolutionary time by plotting the branch length on the x-axis, and the measured or estimated phenotype of each node on the y-axis. Clades defined in (**A**) have been colored accordingly and the paths of 9 key species have been highlighted in bold – for example showing that the last common ancestor of *S. cerevisiae* and *C. albicans* is unlikely to have had a +4 U preference phenotype (last common ancestor of *S. cerevisiae* and *C. albicans* is also indicated by black arrow). *L. starkeyi* = *Lipomyces starkeyi*, *C. fragrans* = *Cephaloascus fragrans*, *E. gossypii* = *Eremothecium gossypii*.

The online version of this article includes the following figure supplement(s) for figure 2:

**Figure supplement 1.** Interspecies association mapping pipeline.

**Figure supplement 2.** Variation in 5'SS splicing signal sequence preference phenotypes across Saccharomycotina.

**Figure supplement 3.** Variation in 5'SS splicing signal sequence preference phenotypes across Saccharomycotina.

**Figure supplement 4.** Variation in 3'SS splicing signal sequence preference phenotypes across Saccharomycotina.

## Switches in 5'SS sequence preference are not associated with compensatory changes in U6 or U1 snRNA sequence

Changes in 5'SS sequence preference could be associated with either the changed function of a spliceosomal protein or changes in UsnRNA sequences that interact with 5'SSs. For example, a 5'SS +4 A could make a Watson-Crick base pair with U6 snRNA through a compensatory nucleotide change in the central adenosine of the U6 ACAGA box (*Figure 3A*). Indeed, a change of the corresponding adenosine to uridine is found in U6 snRNA of the unicellular red alga *Cyanidioschyzon merolae* (*Stark*

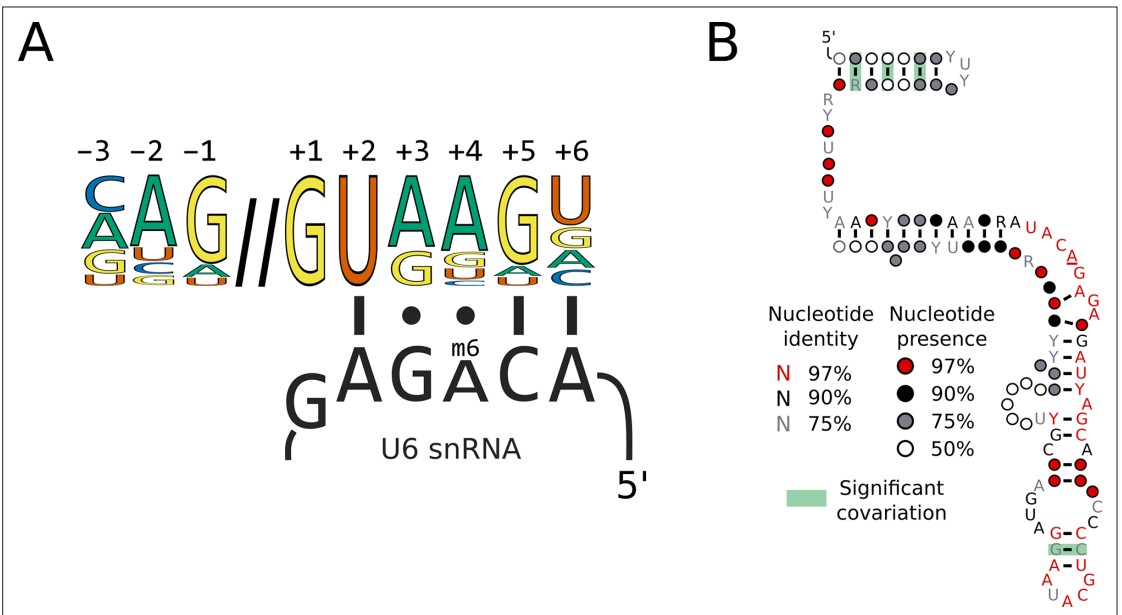

**Figure 3.** Switches in 5'SS sequence preference are not associated with compensatory changes in U6 snRNA sequence. (**A**) Diagram showing the interaction of the U6 snRNA ACAGA box with the approximate human 5'SS consensus sequence. The methylated position of the ACAGA box is located opposite the 5'SS +4 position. (**B**) Conservation of U6 snRNA positions calculated from best U6 snRNA sequences predicted from genomes for 240 species (including 230 Saccharomycotina and 10 outgroups) using Infernal and plotted onto the predicted structure using R-scape and R2R. The predicted structure is derived from the predicted human U6 snRNA structure (*Aoyama et al., 2020*). All the positions in the ACAGA box were 100% conserved across all 240 species.

The online version of this article includes the following figure supplement(s) for figure 3:

**Figure supplement 1.** Switches in 5'SS sequence preference are not associated with compensatory changes in U1 snRNA sequence.

*et al., 2015*; *Wong et al., 2023*). To examine this possibility in our dataset, we predicted a high-confidence U6 snRNA gene in 229 of the 230 Saccharomycotina genomes and 9 of the 10 outgroups' using the software tool Infernal, and a covariance model built using the secondary structural model of human U6 snRNA 5'stem, telestem and internal stem-loop. The remaining two species required manual intervention to annotate the U6 snRNA gene (see Materials and methods). The AC<u>A</u>GA sequence, which is targeted by METTL16, was invariant in all species (*Figure 3B*). Therefore, switches in 5'SS +4 A/U in the Saccharomycotina are not associated with changes in U6 snRNA ACAGA sequences.

A second possibility is that changes in 5'SS +4 preference could be compensated by changes in the sequence or modification of the interacting position of the U1 snRNA (*Figure 3—figure supplement 1A*). During canonical recognition of 5'SSs by U1 snRNA, the 5'SS +4 position interacts with position 5 of the U1 snRNA in the U1/5'SS helix, which is a pseudouridine ($\Psi_5$) in both humans and *S. cerevisiae*. A change in nucleotide identity at this position of U1 snRNA, for example from pseudouridine to adenosine, could compensate a change in 5'SS +4 preference from adenosine to uridine. The U1 snRNA is larger and highly diverged in *Saccharomycetaceae* compared to metazoans, making computational prediction more challenging. However, using a covariance model of the 5'SS interacting site and stem 1 of the U1 snRNA structure, we were able to identify U1 snRNA candidates from 227 of the 230 Saccharomycotina genomes and 9 of the 10 outgroups. The AC<u>U</u>UACC sequence of the 5'SS interacting positions of U1 snRNA was invariant in all species (*Figure 3—figure supplement 1B*).

Overall, we conclude that switches in A/U sequence preference at the 5'SS +4 position in Saccharomycotina is not associated with compensatory sequence changes in the interacting positions of either the U6 or U1 snRNAs.

## Inter-species association mapping links METTL16 to 5'SS +4 sequence preference

Having established variation in the sequence preference of 5'SSs, which could not be explained by compensatory changes in either U1 or U6 snRNA sequence, we next asked if inter-species association mapping could determine whether the presence or absence of a splicing factor or METTL16 ortholog correlated with the 5'SS +4 A/U phenotype.

We first reduced the number of ortholog absences in the dataset caused by gene prediction failures. Multiple sequence alignments of 153 protein orthogroups corresponding to METTL16 and 157 human splicing factors (*Sales-Lee et al., 2021*) were used to generate profile hidden Markov models (pHMMs). The pHMMs were then applied to six-frame translations of Saccharomycotina genomic sequences to rescue missing open reading frames (ORFs) using the software package Hmmer (*Eddy and Pearson, 2011*). Orthogroups were converted into a table of presence/absence variation for each orthogroup in each species, filtering for orthogroups with members present in at least five species and a minimum of two predicted independent loss events in the species tree. These genotypes were then used to perform inter-species association mapping using phylogenetic generalised least squares (PGLS) to identify significant associations with the ratio of 5'SS +4 A to +4 U in each species, whilst controlling for the relatedness of species using distances from the species tree calculated by Orthofinder.

PGLS analysis results in a multiple-testing corrected *p*-value and model coefficient which can be used to examine the correlation of splicing factor presence/absence to the ratio of 5'SS +4 A/U (*Figure 4B*). The strongest and most significant association of the 5'SS +4 A/U phenotype was with the presence or absence of METTL16 orthologs. When METTL16 was present, 5'SS +4 A was more likely to be found, and when METTL16 was absent, 5'SS +4 U was more likely to be found (*Figure 4A*).

METTL16 has been lost from the majority of Saccharomycetaceae and Saccharomycodaceae, and from independent lineages of the CUG-Ser1 and Pichiaceae clades, in association with a switch in 5'SS sequence preference from +4 A to +4 U (*Figure 4C*). The exact number of independent losses in CUG-Ser1 is uncertain due to relatively low confidence splits separating genera within the clade. Two independent subgenera of Saccharomycetaceae, within the Zygosaccharomyces and Eremothecium genera, retain a METTL16 ortholog despite having a strong preference for 5'SS +4 U. Sequence and Alphafold2 structure analysis indicated that the deduced METTL16 protein sequences were complete, incorporating an MTD domain with an intact NPPF catalytic motif (*Figure 4—figure supplement 1A–B*) and a VCR/KA-1-like domain topologically similar to the human METTL16 VCR (*Figure 4—figure supplement 1C*). Association analyses cannot establish causality nor the direction of causation

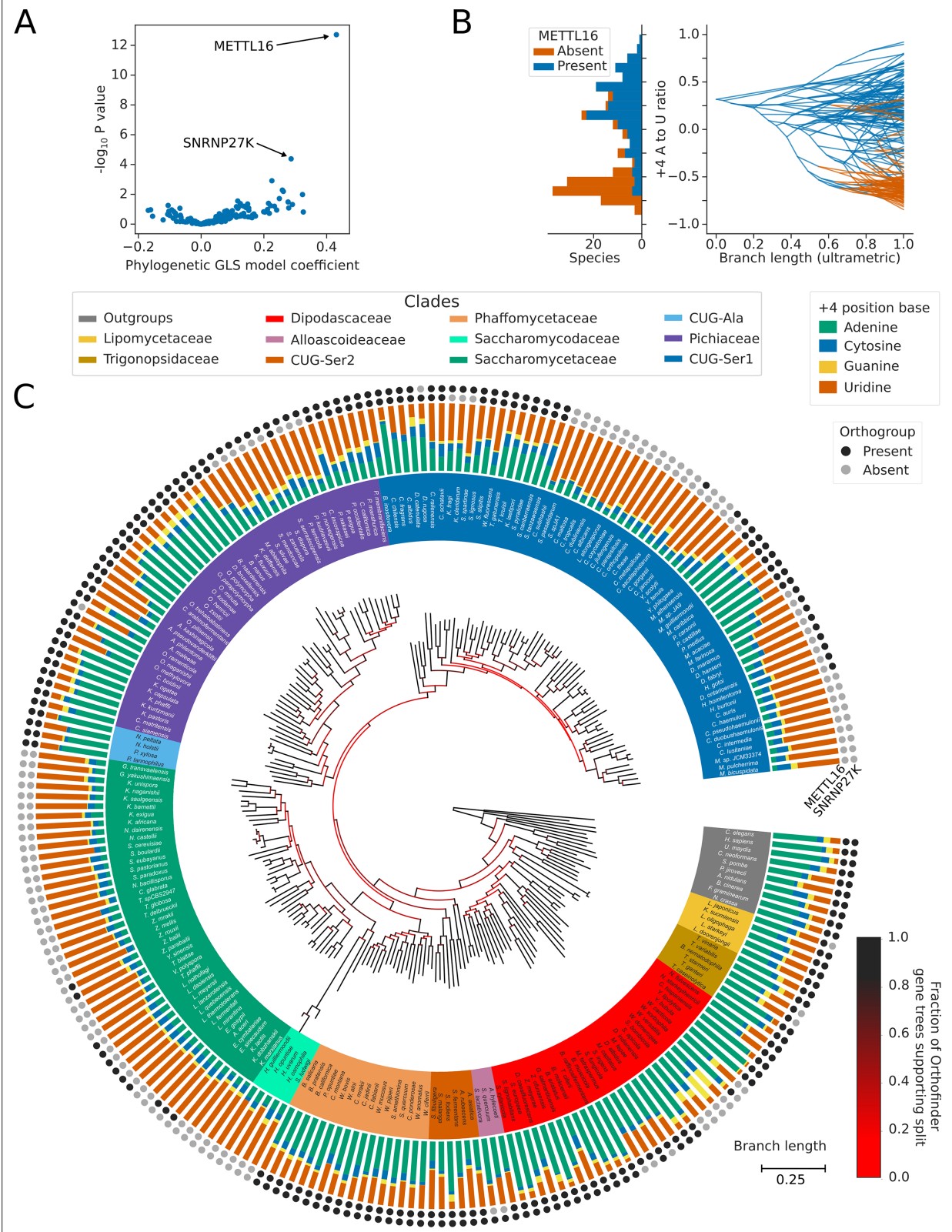

**Figure 4.** Inter-species association mapping links METTL16 to 5'SS +4 sequence preference. (**A**) Volcano plot showing the results of the PGLS analysis of the 5'SS +4 A to U ratio phenotype for orthogroups corresponding to known human splicing factors and METTL16. The magnitude and directionality of the model coefficients (x-axis) are used as a measure of effect size and direction of effect on phenotype, respectively. (**B**) Stacked histogram and traitgram showing the distribution of 5'SS +4 A to U ratio phenotypes for species which retain or lack a METTL16 ortholog. Ancestral loss events shown

*Figure 4 continued on next page*

Figure 4 continued

on the traitgram are estimated using Dollo parsimony. (**C**) Circular tree showing the full phylogenetic relationship between 240 Saccharomycotina species identified using Orthofinder. Bifurcations are colored by confidence, calculated using the number of single-locus gene trees that support the bifurcation in the Orthofinder STAG algorithm (*Emms and Kelly, 2019*; *Emms and Kelly, 2018*). Clades defined in *Figure 2A* have been colored accordingly. Stacked bar charts show the 5′SS +4 position nucleotide fractions of each species identified from conserved introns. Outer circles show the presence (black) or absence (grey) of an ortholog from the orthogroups representing METTL16 (inner ring) and SNRNP27K (outer ring), respectively.

The online version of this article includes the following figure supplement(s) for figure 4:

**Figure supplement 1.** Structural analysis of METTL16 orthologs in the Saccharomycetaceae clade.

in the METTL16/5′SS +4 relationship. If the METTL16 orthologs in Zygosaccharomyces and Eremothecium are functional, expressed as protein, and target U6 snRNA, it may indicate that changes in 5′SS sequence preference in the Saccharomycetaceae occurred prior to METTL16 loss. Overall, we conclude that inter-species association mapping can link METTL16 with the sequence preference of a single nucleotide position in introns.

## Inter-species association mapping links SNRNP27K Methionine 141 to 5′SS sequence preference

Of the splicing factor orthogroups that we analysed, the METTL16 orthogroup was correlated most strongly to 5′SS +4 nucleotide identity in Saccharomycotina (*Figure 4A*). Since this finding is consistent with the outcomes of transcriptome analysis of *Arabidopsis* and *S. pombe* METTL16 ortholog mutants (*Ishigami et al., 2021*; *Parker et al., 2022*) and the cryo-EM structures of *S. cerevisiae* and human spliceosomes (*Bertram et al., 2017*; *Charenton et al., 2019*; *Parker et al., 2022*; *Wan et al., 2019*), we next asked if any of the other correlated orthogroups might provide new insight into the evolution of splice site recognition.

After METTL16, the next most significant association with the 5′SS +4 A/U phenotype was with the orthogroup containing the human spliceosomal protein SNRNP27K (*Figure 4A*). We found that the absence of a SNRNP27K ortholog was associated with an increased preference for 5′SS +4 U in the Saccharomycotina (*Figure 5A*). Although many species that lack SNRNP27K also lack a METTL16 ortholog (*Figure 4C*), we found that the association of SNRNP27K with the 5′SS +4 A/U phenotype was significant even when controlling for the presence or absence of METTL16 (PGLS *P*=0.011). In comparison, the next most significant orthogroup, corresponding to orthologs of U2A′, was not significant when also controlling for the presence or absence of METTL16 and SNRNP27K (PGLS *P*=0.35). In species that lack METTL16, the preference for 5′SS +4 U is even stronger if those species also lack SNRNP27K, demonstrating that the effect of SNRNP27K on the 5′SS +4 A/U phenotype is not wholly explained by correlation with METTL16 (*Figure 5B*). SNRNP27K is a mostly disordered spliceosomal protein with SR-rich regions at the N-terminus and a conserved C-terminal domain (*Fetzer, 1997*; *Zahler et al., 2018*). The C-terminal domain of SNRNP27K is found near the flexible ACAGA sequence of U6 snRNA in cryo-EM structures of human pre-B spliceosomes.

We next asked if inter-species association mapping could be used to examine not just the presence/absence of SNRNP27K, but functional variation in amino acid sequence in the conserved C-terminal domain. The curated profile hidden Markov model (pHMM) of the SNRNP27K C-terminus from Pfam (*Paysan-Lafosse et al., 2023*) was used to generate a consensus sequence for this region of the protein. The predicted protein sequences from 158 Saccharomycotina species that retain a SNRNP27K ortholog were aligned to the pHMM using the software tool Hmmer (*Eddy and Pearson, 2011*) and this alignment was used to generate a table capturing deviation from the consensus sequence in each species (*Figure 5—figure supplement 1A*). We then used PGLS to test whether there was an association with the 5′SS +4 A/U phenotype. The model position corresponding to Methionine 141 (M141) of human SNRNP27K had the strongest association with the 5′SS +4 A/U phenotype (*Figure 5C*, *Figure 5—figure supplement 1B*). In cryo-EM structures of human pre-B spliceosome complexes, SNRNP27K M141 is located within the U-shaped loop of the C-terminal domain close to the U6 snRNA AC**A**GA box, which is in a flexible orientation at this stage (*Charenton et al., 2019*; *Figure 5D*). SNRNP27K is not detectable in the subsequent spliceosomal B complex stage, where the U6/5′SS helix has formed. However, by superimposing the structures of PRP8 from the pre-B and B complexes as a common reference point, it is clear that Met141 of SNRNP27K is particularly close to the space that becomes occupied by U6 snRNA $m^6A_{43}$ (*Figure 5E*). Significantly, a mutation in

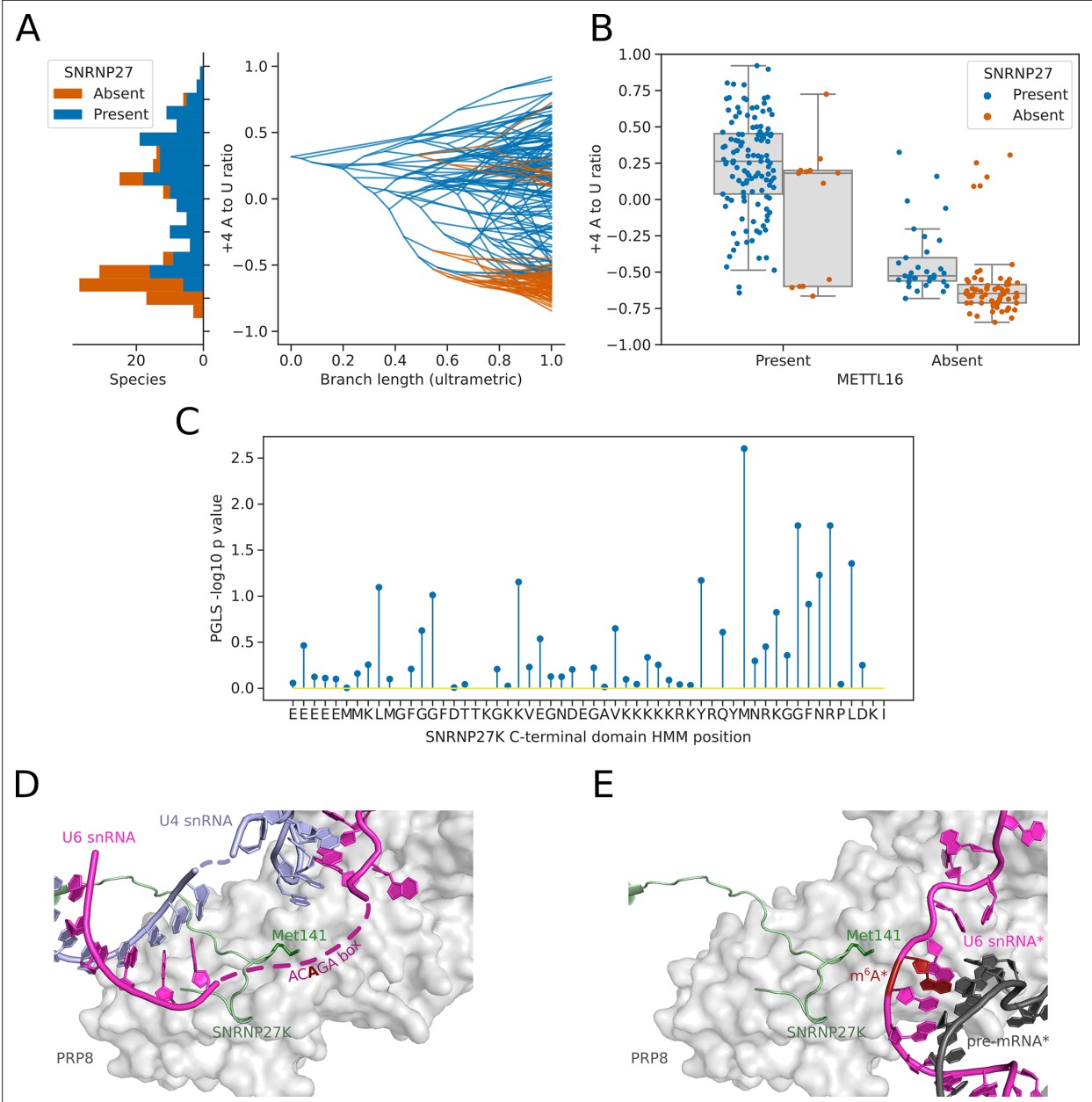

**Figure 5.** Inter-species association mapping links SNRNP27K Methionine 141–5'SS sequence preference. (**A**) Stacked histogram and traitgram showing the distribution of 5'SS +4 A to U ratio phenotypes for species which retain or lack a SNRNP27K ortholog. Ancestral loss events shown on the traitgram are estimated using Dollo parsimony. (**B**) Boxplots showing the distribution of 5'SS +4 A to U ratio phenotypes for species retaining or lacking a SNRNP27K ortholog, whilst also controlling for the presence or absence of a METTL16 ortholog. (**C**) Stem plot showing the association of individual conserved positions in the C-terminus of SNRNP27K with the 5'SS +4 A to U ratio phenotype in 158 species retaining a SNRNP27K ortholog. The most strongly associated position is Met141, which is in extreme proximity to the ACAGA box in the pre-B complex. (**D**) Cryo-EM snapshot of the C-terminus of SNRNP27K in the pre-B complex, in close proximity to the region occupied by the flexible ACAGA box of the U6 snRNA. The surface of PRP8 is shown in grey. (**E**) Overlayed Cryo-EM snapshots showing the position of SNRNP27K in the pre-B complex, relative to the position of the ordered U6/5'SS helix in the B complex. Positioning of the components labelled with asterisks was determined relative to the position of pre-B complex PRP8 by the superimposition of the PRP8 structures from the pre-B and B complexes.

The online version of this article includes the following figure supplement(s) for figure 5:

**Figure supplement 1.** Sequence analysis of the SNRNP27K C-terminal domain.

the corresponding methionine residue of the *C. elegans* SNRNP27K ortholog SNRP-27 to threonine (M141T) was identified in a screen designed to reveal factors that modulate splicing fidelity (*Zahler et al., 2018*). The SNRP-27 M141T mutation caused a shift in 5′SS selection away from 5′SSs that had +4 A, to alternative 5′SSs that did not (*Zahler et al., 2018*). Therefore, orthogonal human spliceosome cryo-EM structures and *C. elegans* mutant RNA-sequencing data are consistent with inter-species association mapping linking SNRNP27K to 5′SS sequence selection.

## Intron number correlates with 5′SS U5 and U6 snRNA interaction potential in species with 5′SS +4A sequence preference

When we previously characterised *Arabidopsis fio1* mutants defective in U6 snRNA m⁶A modification, we found a global switch away from the selection of 5′SSs with +4 A to 5′SSs that not only had +4 U but also a stronger interaction potential with U5 snRNA loop 1 (*Parker et al., 2022*). This led us to investigate the architecture of annotated 5′SSs in genome sequences with respect to U5 and U6 snRNA base-pairing and we showed that there is a negative correlation between 5′SS U5 and U6 snRNA interaction potentials. Hence, two major classes of U2-dependent 5′SSs are found in plants and metazoans (*Parker et al., 2022*). The existence of two major classes of 5′SS may provide a mechanism for regulatable alternative splicing. Given that many Saccharomycotina species have undergone widespread intron loss and have rare or absent alternative splicing, we asked if a negative correlation between 5′SS U5 and U6 snRNA interaction potentials is found in these species.

Position-specific scoring matrices for the 5′SS –3 to –1 and +3 to+7 positions were used to assess the interaction potential with U5 and U6 snRNA, respectively (Figure A). We used the Spearman rank method to assess the level of correlation or anti-correlation between these U5 and U6 snRNA interaction potentials - we refer to this metric as *U5/6ρ* (*Figure 6A*). The number of conserved introns identified in each species was used as an estimate of intron number. The number of conserved introns and *U5/6ρ* were then compared using PGLS to control for phylogenetic structure. This analysis revealed that for species with fewer introns (and therefore, likely reduced splicing complexity), there was an increase in *U5/6ρ*, indicating weaker anti-correlation of U5 and U6 snRNA interaction potentials (*Figure 6B*). For species with approximately 500 or fewer conserved introns, *U5/6ρ* was more likely to be positive (*Figure 6B*). This suggests that as splicing complexity is reduced, anti-correlated U5 and U6 snRNA interaction potentials are lost. Therefore, the two classes of 5′SS found in plants and metazoans are not found in Saccharomycotina species with reduced splicing complexity.

The relationship between intron number and *U5/6ρ* was different for intron-rich species compared with intron-poor species. *U5/6ρ* is strongly negatively correlated with intron number amongst species with more than approximately 300 conserved introns, but weakly correlated amongst species with fewer conserved introns (*Figure 6B*). Most species with a 5′SS +4 U preference were intron-poor, and the median *U5/6ρ* for species with an overall preference for 5′SS +4 U was positive, with relatively few species showing a negative *U5/6ρ*. This suggests that a switch in preference towards 5′SS +4 U may accompany, or occur after, an overall reduction in intron number, splicing complexity and/or loss of alternative splicing.

*U5/6ρ* was tested as a splicing phenotype with inter-species association mapping, but no significant correlations were found after multiple testing correction (*Figure 6—figure supplement 1*). Overall, we conclude that a preference for 5′SS +4 U is a feature of species with low intron numbers and likely low splicing complexity. In contrast, a preference for 5′SS +4 A is a feature of diverse species that exhibit a correlation between intron number and the presence of two major classes of 5′SS that have anti-correlated U5 and U6 snRNA interaction potentials.

## Discussion

The last eukaryotic common ancestor likely had higher intron density and more complex spliceosomes than is observed in developmentally simple eukaryotes like the saccharomycetous yeasts (*Irimia et al., 2007*; *Jeffares et al., 2006*; *Sales-Lee et al., 2021*). The processes that led to splicing simplification in these clades are unclear. We used phylogenetic analysis to reveal a high level of variation at the +4 position of 5′SSs across the Saccharomycotina. Inter-species association mapping demonstrates that variation at the 5′SS +4 position is correlated with the presence or absence of the U6 snRNA N6-methyladenosine methyltransferase, METTL16: when METTL16 is present, a 5′SS +4 A is preferred, but

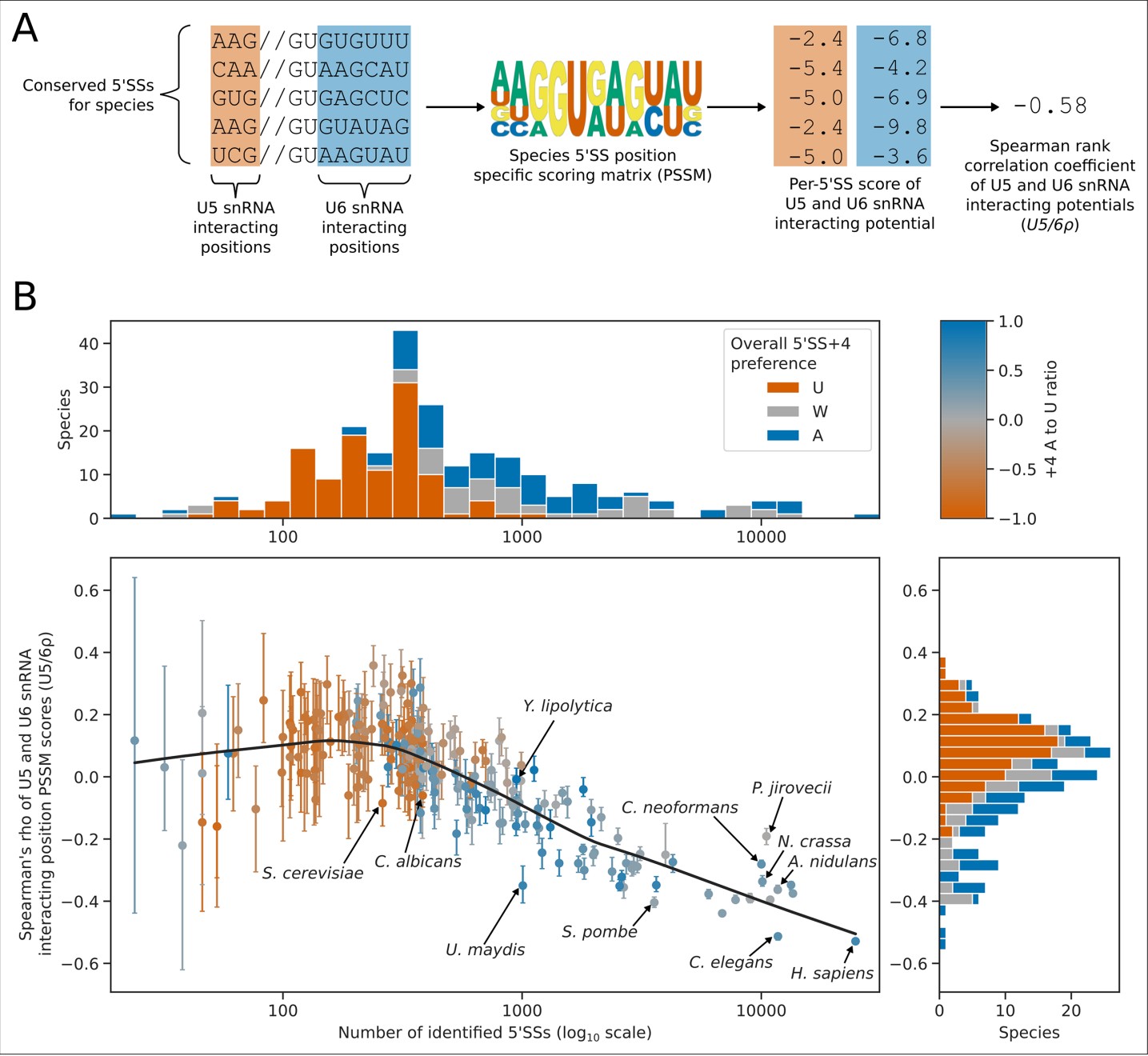

**Figure 6.** 5'SS +4 sequence preference interacts with intron number and U5/U6 interaction strength patterns. (**A**) Diagram showing the calculation of the *U5/6ρ* metric. For each species, all conserved 5'SSs were used to generate a log$_2$ transformed position specific scoring matrix (PSSM) representing the consensus 5'SS sequence in that species. This PSSM was then used to score how well each individual 5'SS matched the consensus at the U5 and U6 snRNA interacting positions. These scores were used as a measure of U5 and U6 snRNA interaction potential for each 5'SS. Per-5'SS U5 and U6 snRNA interaction potentials were then correlated using the Spearman rank method to give a correlation coefficient for each species. This metric is referred to as *U5/6ρ*. In the example given, the species has a negative *U5/6ρ* indicating an overall anti-correlation between the U5 and U6 snRNA interaction potentials of individual 5'SSs. (**B**) Scatterplot with marginal histograms showing the relationship between the number of conserved introns (scatterplot x-axis), the correlation of U5 and U6snRNA interaction potentials (*U5/6ρ*, scatterplot y-axis), and 5'SS +4 nucleotide preference (color). Marginal histograms show the distribution of conserved intron size (top margin) and *U5/6ρ* (right margin) amongst species. For scatterplot points, 5'SS +4 A to U ratio is shaded using the color map shown on the top right. For marginal histograms, 5'SS +4 preference is discretised into an overall preference for either A (A to U ratio >0.2, blue), U (A to U ratio <= –0.2, orange), or no overall preference (–0.2<A to U ratio ≤ 0.2, grey). Confidence intervals for the *U5/6ρ* metric were obtained by bootstrapped resampling of conserved 5'SSs for each species before performing the calculations described in (**A**). The lowess (locally weighted scatterplot smoothing) regression line indicates a strong negative relationship between *U5/6ρ* and conserved intron number in intron rich species, compared to a weak relationship for intron poor species.

*Figure 6 continued on next page*

*Figure 6 continued*

The online version of this article includes the following figure supplement(s) for figure 6:

**Figure supplement 1.** Interspecies association mapping of the *U5/6ρ* phenotype.

when METTL16 is absent, a 5'SS +4 U is preferred. We also show that a C-terminal domain that increases METTL16 specificity for the U6 snRNA (*Aoyama et al., 2020*) is ancestral to most eukaryotes. Together, these results suggest that the primary conserved role of METTL16 is to modify the ACAGA box of the U6 snRNA.

We identified a further correlation between 5'SS +4 preference and the spliceosomal protein SNRNP27K. In this case, we could also identify an association with the methionine residue corresponding to position 141 of human SNRNP27K in the conserved C-terminal domain of SNRNP27K orthologs. This finding is consistent with global transcriptome analysis of *C. elegans* SNRP-27 M141T mutants, which show a switch away from using 5'SS +4 A (*Zahler et al., 2018*). Therefore, inter-species association mapping can link the function of individual proteins to switches in 5'SS sequence preference found between species.

## How do interactions at the 5'SS +4 position affect splice site selection?

Cryo-EM structures of human B complex spliceosomes show that U6 snRNA $m^6A_{43}$ forms a trans Hoogsteen sugar edge interaction with 5'SS +4 A (*Bertram et al., 2017*; *Parker et al., 2022*). The U6 snRNA $m^6A_{43}$ may allow 5'SS +4 preferences to become more flexible because biophysical data indicates that $m^6A$ stabilises A:A base pairs whilst destabilising A:U base pairs (*Kierzek and Kierzek, 2003*; *Roost et al., 2015*). Consistently, in the absence of METTL16 and U6 snRNA $m^6A$ modification, RNA sequencing data shows a switch in preference from 5'SS +4 A to 5'SS +4 U (*Ishigami et al., 2021*; *Parker et al., 2022*). Inter-species association mapping demonstrates that these findings from mutants within species are replicated at evolutionary timescales in different Saccharomycotina species.

Cryo-EM structures of *S. cerevisiae* spliceosomes show that a Watson-Crick base pair is made between 5'SS +4 U and the central non-$m^6A$-modified adenosine of the U6 snRNA ACAGA box (*Wan et al., 2019*). Biophysical data indicate that although an $m^6A$:U base pair can form in a duplex, the methylamino group rotates from a *syn* geometry on the Watson–Crick face to a higher energy anti-conformation, positioning the methyl group in the major groove (*Roost et al., 2015*). As a result, $m^6A$ has a destabilising effect on A:U base pairs in short RNA helices (*Kierzek and Kierzek, 2003*; *Roost et al., 2015*). Consistent with this, the splicing efficiency of introns that have 5'SS +4 U is increased in *Arabidopsis fio1* (*Parker et al., 2022*) and *S. pombe mtl16Δ* (*Ishigami et al., 2021*) mutants. These findings are consistent with the inter-species association mapping results which link the absence of METTL16 to a preference for 5'SS +4 U. Therefore, $m^6A$ modification of U6 snRNA appears generally incompatible with a preferred selection of 5'SS +4 U sequences.

Inter-species association mapping indicated that the correlation between SNRNP27K presence/absence and the 5'SS +4 A/U phenotype persists in species lacking METTL16, suggesting that retention of an SNRNP27K ortholog allows more tolerance of 5'SS +4 A in the absence of METTL16 and U6 snRNA $m^6A$ modification. However, relatively few species of Saccharomycotina have lost SNRNP27K but retained a METTL16 ortholog, hinting that the role of SNRNP27K might be necessary when U6 snRNA is $m^6A$-modified. SNRNP27K is detected in pre-B spliceosomes close to the U6 snRNA AC**A**GA sequence, which is in a flexible configuration prior to the transfer of the 5'SS from U1 snRNA (*Charenton et al., 2019*). Met141 of SNRNP27K is located close to the position where the $m^6A_{43}$:5'SS +4 A pairing of the U6/5'SS helix forms in B complexes, although SNRNP27K has left the spliceosome by this stage. Additional cryo-EM snapshots of intermediate spliceosome states are required to determine the conformational and compositional changes that occur between the pre-B and B complexes. However, it seems likely that SNRNP27K stabilises the formation of the U6/5'SS helix product state by chaperoning the arrangement of the flexible U6 snRNA AC**A**GA sequence for 5'SS docking.

## Why does 5'SS +4 sequence preference switch from A to U?

Our analysis suggests that METTL16 and 5'SSs with +4 A sequence preference are features of early eukaryotes. In contrast, 5'SSs with a +4 U preference appear to have evolved later and on multiple independent occasions in saccharomycetous yeasts, a group of species that have simplified

developmental complexity and spend most or all of their life cycles in a unicellular form (*Nagy et al., 2014*). A series of changes are associated with species that have reduced multicellularity, including reductions in intron density and the evolution of more invariant 5'SS sequences (*Irimia et al., 2007*; *Lim et al., 2021*; *Sales-Lee et al., 2021*; *Schwartz et al., 2008*). We demonstrate that a switch to 5'SS +4 U is a feature of these changes in Saccharomycotina.

A key question arising from evolutionary differences in 5'SS sequence preferences is why a switch between 5'SS +4 A/U is found. We previously reported that there are two major classes of 5'SS in many eukaryotes, defined by their anti-correlated interaction potential with U5 and U6 snRNA (*Kenny et al., 2022*; *Parker et al., 2022*). This variation in 5'SS composition may provide regulatory potential in alternative splice site selection because pairs of alternative 5'SSs tend to be from the two opposing classes (*Parker et al., 2022*). Our analysis demonstrates that the anti-correlation of U5 and U6 snRNA interaction potentials is weaker, and even becomes positive, in species that have a reduced number of conserved introns. This may be due to the reduced importance of alternative splicing as a regulatory mechanism of gene expression in these species. Furthermore, we show that major switches in 5'SS +4 preference have occurred almost exclusively in Saccharomycotina species that lack a strong anti-correlation between U5 and U6 snRNA interaction potentials. We hypothesise that U6 snRNA m⁶A modification might provide plasticity to 5'SS selection, either through the modulation of U6 snRNA m⁶A levels directly, or through protein chaperones such as SNRNP27K. This plasticity could be harnessed either within individuals by means of regulatory networks, or in populations due to natural variation in the relevant splicing factors (*Price et al., 2020*; *Sasaki et al., 2015*). In species that have lost splicing complexity, U6 snRNA m⁶A modification may have become detrimental, since plasticity intrinsically means weaker signals that are more prone to cryptic or mis-splicing. Similar reasoning has previously been suggested to explain the link between reduced intron density and more invariant 5'SS sequences (*Irimia et al., 2007*). Switches in 5'SS +4 preference from A to U may therefore represent one of the final stages of splicing simplification.

## What is the order of events directing evolutionary change in splicing signal phenotypes?

Inter-species association mapping reveals genotypes that correlate with a phenotype but cannot necessarily prove causation. Consequently, it is not possible to reconstruct the order of evolutionary changes we detect. However, different scenarios that might explain these findings can be considered. For example, sudden loss of METTL16 might cause an urgent necessity to change 5'SS sequence preferences. This possibility seems unlikely as such rapid change without widespread corresponding 5'SS changes would likely impose a high fitness cost. Alternatively, change in 5'SS sequence from +4 A to +4 U preference could occur first, driven by some other selective pressure such as widespread intron loss, until there is no longer a benefit to retaining the METTL16 gene. This scenario is consistent with the fact that we could detect the METTL16 gene in *Zygosaccharomyces* and *Eremothecium* species that have altered their 5'SS +4 preference to a U. However, biophysical data of m⁶A:U interactions and global splicing analyses of METTL16 ortholog mutants indicate that the co-occurrence of METTL16 and +4 U are generally incompatible (*Ishigami et al., 2021*; *Kierzek and Kierzek, 2003*; *Parker et al., 2022*; *Roost et al., 2015*). Instead, it is possible that gradual changes in the expression or catalytic efficiency of METTL16 could reduce the stoichiometry of U6 snRNA m⁶A-modification, permitting a gradual change in 5'SS +4 sequence preference, until complete loss of the METTL16 gene no longer imposes a major fitness cost. Future work might examine these scenarios by determining whether the METTL16 genes detected in *Zygosaccharomyces* and *Eremothecium* species are expressed and functional.

## Inter-species association mapping is under-utilised

Phylogenetics has played important roles in developing a mechanistic understanding of the RNA components of the spliceosome (*Guthrie, 2010*). Our analysis demonstrates that inter-species association mapping can be a powerful approach to elucidate molecular interactions controlling splicing. The 5'SS +4 preference phenotype studied here appears to be an example of a relatively simple trait that is predominantly associated with the presence or absence of a small number of genes, i.e., METTL16 and SNRNP27K. Other splicing phenotypes are likely to be more complex, involving smaller contributions from numerous genes as well as associations at the level of protein domains or individual amino

acids. These issues may explain why we were unable to identify any candidate orthogroups associated with the U5/6ρ phenotype. As in GWAS, the solution to understanding complex traits may be larger studies (*Tam et al., 2019*). However, analysing increasing numbers of genomes will require scalable tools for high-quality gene annotation and ortholog clustering (for example, using gene synteny relationships). Since splicing occurs on RNA, transcriptomic data will be valuable for measuring phenotypes such as rates of alternative splicing, as well as aiding the annotation of protein-coding genes. The increasing availability of tree-of-life-scale genomics data from ambitious sequencing projects that aim to catalogue the genomes of all eukaryotic life on Earth (*Darwin Tree of Life Project Consortium, 2022*; *Lewin et al., 2018*) will provide resources to improve the quality and quantity of data to broaden the future application of inter-species association mapping (*Smith et al., 2020*).

# Materials and methods

## METTL16 structural bioinformatics

Orthologous METTL16 sequences and Alphafold2 structural predictions were manually curated from UniProt using the EBI phmmer server (*Potter et al., 2018*; *Bateman et al., 2023*; *Varadi et al., 2022*). Alphafold2 structural predictions were segmented into domains using network analysis as previously described (*Oeffner et al., 2022*). For each pair of residues $i$ and $j$ with minimum predicted local distance difference test (pLDDT) values of greater than 70 and a pairwise alignment error (PAE) of less than 5, we connected the residues in a graph using an edge weight of $PAE_{ij}^{-1}$. Residues were then clustered into domains of connected amino acids using the greedy modularity maximisation method implemented in networkx (*Hagberg et al., 2008*). Domains with fewer than 20 residues were discarded.

To assess the topological similarity of the discovered domains with human METTL16 MTD and VCR/KA-1 domains, the crystal structures of METTL16 MTD (PDB: 6B91; *Ruszkowska et al., 2018*), METTL16 VCR (PDB: 6M1U; *Aoyama et al., 2020*) and TUT1 KA-1 (PDB: 5WU5; *Yamashita et al., 2017*) were downloaded from the Protein Data Bank (*Burley et al., 2019*). Pairwise TM-scores were calculated for all pairs of experimentally determined and predicted domains using TM-align (*Zhang and Skolnick, 2005*). Domains with a TM-score of 0.45 or greater compared to human METTL16 MTD (6B91) or VCR (6M1U) were assigned to the MTD-like and KA-1/VCR-like clusters, respectively. Domains with TM-scores of 0.45 or greater compared to both METTL16 MTD and VCR domains were assigned to the MTD +KA-1/VCR-like cluster. Multiple structure alignments and overall cluster TM-scores were generated using US-align (*Zhang et al., 2022*). Figures were drawn using matplotlib, PyMol, ETE 3, and Jalview (*Huerta-Cepas et al., 2016*; *Hunter, 2007*; *Procter et al., 2021*; *Schrödinger, 2015*).

## Genome annotation, orthogroup clustering and 5'SS+4 phenotype estimation

The representative genomes of 230 Saccharomycotina species and 10 outgroups were downloaded from NCBI. Where possible, the RefSeq genome was used, however for some species only GenBank versions were available. See *Supplementary file 1* for the full list of genome metadata. For the 10 well-annotated outgroups and 3 reference Saccharomycotina species (*S. cerevisiae*, *C. albicans*, *Y. lipolytica*), the publicly available gene annotations were also downloaded in GTF format. For all other species, genomes were cleaned (by removal of short repetitive contigs), repeat masked, and reannotated using the Funannotate pipeline using default settings (with internal processing using Minimap2, Tantan, BUSCO, AUGUSTUS, GeneMark, GlimmerHMM and SNAP) (*Borodovsky and Lomsadze, 2011*; *Frith, 2011*; *Korf, 2004*; *Li, 2018*; *Majoros et al., 2004*; *Palmer and Stajich, 2020*; *Seppey et al., 2019*; *Stanke et al., 2006*). Seed species for gene finding were chosen based on known phylogenetic clades: Lipomycetaceae, Trigonopsidaceae and Dipodascaceae species were seeded using *Y. lipolytica*. Alloascoideaceae, Sporopachydermia, CUG-Ala, Pichiaceae and CUG-Ser1 species were seeded using *C. albicans*. CUG-Ser2, Phaffomycetaceae, Saccharomycodaceae and Saccharomycetaceae species were seeded using *S. cerevisiae*.

After gene prediction, protein sequences were clustered into orthogroups using Orthofinder version 2.5.4 (*Emms and Kelly, 2019*) with DIAMOND for pairwise proteome comparisons and DendroBLAST for gene tree inference (*Buchfink et al., 2021*; *Kelly and Maini, 2013*). Orthofinder also produces a species tree using the STAG method (*Emms and Kelly, 2018*). We found that many species initially

had spurious ortholog absences due to gene prediction failure. To address this for the 158 splicing factors being examined, we used the human orthologs (*Sales-Lee et al., 2021*) to identify relevant orthogroups and use the sequences from these to build profile hidden Markov models (pHMM) using MAFFT and hmmer (*Eddy and Pearson, 2011*; *Katoh and Standley, 2013*). Score thresholds for each pHMM were determined by aligning the input sequences to the pHMM and selecting the 2.5% percentile (i.e. the score threshold that would recover 97.5% of the input sequences). These were then used to search six-frame translations generated from genome sequences to recover missing orthologs. Finally, the recovered sequences were re-clustered using Orthofinder to generate refined splicing factor orthogroups which were annotated using the human orthologs present within them.

We found that initial intron predictions generated by the Funannotate pipeline contained many false positive introns that negatively affected the estimation of splicing signal motifs. We reasoned that since most genuine introns are conserved across multiple species, and incorrectly annotated introns are unlikely to be in the same position in multiple species by chance, then identifying conserved introns would be a suitable method to reduce the false positive rate. To do this, we took the protein sequences for the initial orthogroups (generated from gene predictions rather than six-frame translations of splicing factors) and generated multiple sequence alignments using MAFFT. Very large orthogroups (with an average of more than 2 orthologs per species) were skipped as a time-saving heuristic. Once multiple sequence alignments were generated, we filtered for introns that were conserved in the same alignment column and frame in at least two species. This significantly improved the information content of splicing motifs at the 5' and 3'SSs of introns, meaning that 5'SS +4 preference phenotypes could be estimated. Since the number of conserved introns per species varied by several orders of magnitude, we also used bootstrapped resampling of introns to estimate the 95% confidence intervals on the 5'SS +4 preference phenotypes.

## Phylogenetic generalised least squares analysis

After a genotype (in the form of orthogroups), phenotype (5'SS +4 preference) and phylogenetic relationships (species tree) had been collected, we were able to proceed with PGLS analysis. Phylogenetic dependence was estimated from the species tree by generating a variance-covariance matrix. Variances were set as the distance from the root to species leaf, and covariances were set as the distance from the root to the last common ancestor of the two species (*de Villemereuil et al., 2012*). For each orthogroup, the data were binarized to give the presence/absence of an ortholog for each species. Orthogroups were filtered using Dollo parsimony to retain those predicted to have undergone at least two independent loss events (*Farris, 1977*; *Rogozin et al., 2003*). PGLS analysis was then conducted for each orthogroup using the GLS method from statsmodels, controlled with the covariance matrix (*Seabold and Perktold, 2010*). To account for uncertainty in the 5'SS +4 phenotype, bootstrapped 5'SS +4 A to U ratios estimated using resampling of conserved introns were used to perform 100 independent PGLS tests per orthogroup, which were then combined to give a unified model coefficient (with 95% confidence intervals) and *p* value using the GLS pooling method from the statsmodels implementation of MICE (*Seabold and Perktold, 2010*). p Values for each orthogroup were then corrected for multiple testing using the Benjamini-Hochberg method.

For the individual amino acid PGLS analyses of the SNRNP27K C-terminus, genotypes used for testing were prepared as follows: the curated pHMM of the SNRNP27K C-terminus was downloaded from Pfam (PF08648). SNRNP27K orthologs were aligned to the pHMM using hmmalign (*Eddy and Pearson, 2011*), discarding insertions to the model, and the highest-scoring ortholog for each species was selected. The consensus sequence of the pHMM model was collected using hmmemit (*Eddy and Pearson, 2011*). For each ortholog sequence, at each position in the model, a score was assigned by looking up the pairwise similarity scores of the aligned ortholog amino acid and the consensus amino acid in the BLOSUM62 matrix. Scores for each column were standardised by subtracting the mean score and dividing by the standard deviation. The resulting table was used to test each conserved position separately, using the same bootstrapped PGLS method described above.

## U6 and U1 snRNA conservation analysis

A seed alignment of U6 snRNA sequences from the 10 outgroups and 3 reference Saccharomycotina genomes was generated using U6 snRNA sequences manually curated from RNACentral (*Sweeney et al., 2021*) and the secondary structural model of the human U6 snRNA (*Aoyama et al., 2020*;

*Montemayor et al., 2014*). Sequences were aligned to the sequence and secondary structural model of human U6 snRNA using Infernal cmalign to create the seed alignment (*Nawrocki and Eddy, 2013*). This seed alignment was then used to create a covariance model that was used to search the genomes of the 240 species using Infernal cmsearch (*Nawrocki and Eddy, 2013*). We were unable to identify a U6 snRNA gene in the representative genome of *Komagataella phaffii* using this method. This appears to be the result of an assembly error since a U6 snRNA sequence could be manually recovered from other published *K. phaffii* genomes by BLAST search with the U6 snRNA sequence identified in *K. pastoris*. The U6 snRNA of *S. pombe* was also not detected by this approach because it is interrupted by a pre-mRNA-like intron (*Potashkin and Frendewey, 1989*). Therefore, we manually annotated the *S. pombe* U6 snRNA sequence using data from RNAcentral (*Sweeney et al., 2021*). The highest scoring hit per species was selected as the representative U6 snRNA ortholog for each species and then aligned to the human U6 snRNA structure using cmalign. Finally, R-scape and R2R were used to plot conservation and covariation relationships onto the consensus U6 snRNA structure (*Rivas et al., 2017*; *Weinberg and Breaker, 2011*).

An initial model of U1 stem 1 was created by manually truncating the covariance model of metazoan U1 snRNA from Rfam to include only the 5' end and U1 stem 1 region. The sequences of U1 snRNA from 7 outgroups and 15 species of Saccharomycotina were collected either from RNAcentral or manual curation from genomic sequences using the U1 stem 1 covariance model and aligned using the initial covariance model to create a seed alignment. This seed alignment was then used to create a final covariance model that was used to search the genomes of all 240 species using the same procedures as described above for U6 snRNA. We were unable to detect U1 snRNA genes from *Cryptococcus neoformans*, *Hanseniaspora uvarum*, *Hanseniaspora guilliermondii*, or *Saturnispora silvae* using this approach.

## Phylogenetic trees and traitgrams

Ancestral character estimations for the 5'SS +4 A to U ratio (used for traitograms) were estimated using the ACE method from the R package Ape (*Paradis et al., 2004*). Ancestral splicing motif nucleotide fractions (i.e. the stacked bar preferences in *Figure 2A*) were estimated using Sankoff parsimony using a previously described method (*Schwartz et al., 2008*), with nucleotide fractions discretised to multiples of 0.05. Phlyogenetic trees and traitgrams were drawn using ete3 and matplotlib (*Huerta-Cepas et al., 2016*; *Hunter, 2007*).

## Analysis of cryo-electron microscopy structures

The cryo-EM structures of the human pre-B (PDB: 6QX9; *Charenton et al., 2019*) and B complexes (PDB: 6AHD; *Zhan et al., 2018*) were downloaded from the Protein Data Bank (*Burley et al., 2019*). The pre-B and B complex structures were superimposed with PyMol using the relative positions of the spliceosomal protein PRP8 (*Schrödinger, 2015*).

## Analysis of U5 and U6 snRNA interaction potential correlations

*U5/6ρ* for each species was estimated using conserved introns (as defined in Materials and methods section '*Genome annotation, orthogroup clustering and 5'SS +4 phenotype estimation*'). For each species, conserved introns were used to create a position-specific scoring matrix describing the 5'SS consensus sequence. The –3 to –1 positions of this matrix were used to score the U5 snRNA interacting potential of each 5'SS, whilst the +3 to+7 positions were used to score U6 snRNA interacting potential. The Spearman rank correlation coefficient of these two scores was used to define *U5/6ρ*. Bootstrapped 95% confidence intervals were generated by resampling introns from each species with replacement. The association between *U5/6ρ* and conserved intron number was assessed using PGLS (as described in Materials and methods section 'Phylogenetic generalised least squares analysis').

## Code availability

All pipelines, scripts and notebooks used to generate figures are available from GitHub at github. com/bartongroup/mettl16_phylogenetics (copy archived at *Bartongroup, 2023*).

## Acknowledgements

We thank Alper Akay, Brendan Davies, Carey Metheringham and Korbinian Schneeberger for comments on the manuscript. This work was supported by awards from the BBSRC (BB/V010662/1 to GGS and BB/W007673/1 to GGS and GJB); SMF is a Wellcome Trust and Royal Society Sir Henry Dale Fellow (grant number 220212/Z/20/Z).

## Additional information

### Funding

| Funder | Grant reference number | Author |
|---|---|---|
| Biotechnology and Biological Sciences Research Council | BB/V010662/1 | Gordon G Simpson |
| Biotechnology and Biological Sciences Research Council | BB/W007673/1 | Geoffrey J Barton<br>Gordon G Simpson |
| Wellcome Trust | 220212/Z/20/Z | Sebastian M Fica |

The funders had no role in study design, data collection and interpretation, or the decision to submit the work for publication. For the purpose of Open Access, the authors have applied a CC BY public copyright license to any Author Accepted Manuscript version arising from this submission.

### Author contributions

Matthew T Parker, Conceptualization, Software, Formal analysis, Investigation, Visualization, Methodology, Writing – original draft, Writing – review and editing; Sebastian M Fica, Conceptualization, Formal analysis, Funding acquisition, Writing – review and editing; Geoffrey J Barton, Conceptualization, Supervision, Funding acquisition, Project administration, Writing – review and editing; Gordon G Simpson, Conceptualization, Supervision, Funding acquisition, Writing – original draft, Project administration, Writing – review and editing

### Author ORCIDs

Matthew T Parker ⓘ http://orcid.org/0000-0002-0891-8495
Sebastian M Fica ⓘ http://orcid.org/0000-0001-9186-0361
Geoffrey J Barton ⓘ http://orcid.org/0000-0002-9014-5355
Gordon G Simpson ⓘ https://orcid.org/0000-0001-6744-5889

### Decision letter and Author response

Decision letter https://doi.org/10.7554/eLife.91997.sa1
Author response https://doi.org/10.7554/eLife.91997.sa2

## Additional files

### Supplementary files

• Supplementary file 1. Table containing the 240 genome assemblies used for phylogenetic association mapping. All assemblies were downloaded from NCBI. Where available, RefSeq assemblies were preferred over GenBank.

• MDAR checklist

### Data availability

This study uses the publicly available genome sequences of 240 different species. The details of the previously used data sets corresponding to these 240 genome sequences are reported in *Supplementary file 1*.

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
