## [Editor Report]

The manuscript addresses the ways in which different organisms have evolved pre-messenger RNA systems that are either more or less complex, a question that underlies the evolution of complex organisms and the genome adaptation of simple organisms to their specific environments. This important manuscript provides the underlying molecular mechanisms of how 5' splice site sequence preference may have evolved, with solid structural modeling data in support.

---

## [Decision Letter]

[Editors' note: this paper was reviewed by Review Commons.]

---

## [Author Response]

*General Statements*

We are grateful to the reviewers for the constructive comments designed to improve our manuscript. All three reviewers recognise the broad impact of our study. We have responded to each of the comments they make as detailed below and revised our manuscript accordingly.

We have reworded statements in the Abstract and Discussion about the evolutionary relationship between METTL16 and 5’SS sequence preference, to more carefully reflect what we can currently conclude from the data analysed in our study. We have also updated the Discussion to include a new section that addresses the evolutionary scenarios that could explain these data.We have edited the Summary and Introduction sections to give a more balanced view of the link between splicing complexity and developmental complexity.We have added 5’SS sequence motifs to the x-axis of figure 2B to make the plot clearer.We have created a pruned tree showing the 5’SS motifs of a selection of Saccharomycotina species, which further demonstrates that the changes in 5’SS+4 position preferences seen in *S. cerevisiae* and *C. albicans* are likely to be a result of convergent evolution. We have added this tree as Figure 2 —figure supplement 3.We have altered the way in which we describe and visualise the relationship between conserved intron number and U5/6rho in Figure 6B, to remove the implication that changes in +4 preference, rather than changes in intron number, explain this trend.

Reviewer #1 (Evidence, reproducibility and clarity (Required)):SummaryThe manuscript by Parker et al. addresses the important question of how different organisms have evolved pre-messenger RNA systems that are either more or less complex. This question underlies the evolution of complex organisms and the genome adaptation of simple organisms to their specific environments, so is an important question to answer. This manuscript now provides the underlying molecular mechanisms of how 5' splice site sequence preference may have evolved which is both an interesting and exciting advance for the field.

We thank the reviewer for these kind comments.

Major commentsThis manuscript builds on the previous work from this group where they identified the role of adenosine N6 methylation (m6A) of the U6 small nuclear RNA (snRNA) of the spliceosome by METTL16 as being important for 5' splice site selection. This work led to the speculation that loss of a METTL16 ortholog, or potentially other splicing factors, in some species could contribute to an evolutionary change in 5' splice site sequence preference. Here the authors now use the power of phylogenetics, interspecies association mapping and the available spliceosome structures to provide convincing conclusions that 5' splice site sequence preferences in the extensive number of organisms examined correlate with the presence of the U6 snRNA methyltransferase METTL16 and the splicing factor SNRNP27K.An analysis of METTL16 conservation was first carried out by comparing the METTL16 methyltransferase domain (MTD) in 29 diverse eukaryotic species. All the METTL16 orthologs were found to have either one or two globular domains. Three domain types were identified and compared in detail. What was not clear from this analysis was the functional significance of orthologs having either one or two domains.

We identified several species, including *Drosophila melanogaster*, whose METTL16 orthologs do not contain a VCR domain. However, in this study we do not draw specific conclusions about the functional significance of orthologs having different domain topologies.

In addition, while this analysis provides important new information on the domain structure of METTL16 orthologs, especially where these domains had not been identified previously, the link between this section of the results and the following sections is not that apparent.

We agree that there is a significant difference in approach between the first section of the Results and the following sections. However, we are keen to keep this part of the manuscript because it provides an orthogonal line of evidence suggesting that the ancestral role of METTL16 in eukaryotes is specifically the methylation of U6 snRNA.

Next novel bioinformatics pipelines were developed to compare both introns and orthologous groupings of protein coding genes between 227 Sacchromycotina genomes as well as 13 well-annotated eukaryote genomes. First, the 5' splice site sequence preference was compared and clearly indicates that the +4 position has the greatest variation in preferences within the Sacchromycotina. The ability to now compare a large number of genomes has provided novel information on the evolution of the 5' splice site sequence and the conclusion that there is more complexity to the 5' splice site in fungi that previously recognized. While it is apparent why only the 5' splice site signal was investigated here, with its relationship to the U6 snRNA and METTL16, it seems a shame the other splice site sequences were not analyzed using this novel pipeline. In any case, the complexity of the 5' splice site +4 position now allows, for the first time, interesting interspecies association studies.

We have now included the variance plots for 3’SS motifs (analogous to the 5’SS variance plots shown in Figure 2B) as Figure 2 supplementary figure 4A, and a traitgram for 3’SS -3C to U ratio as Figure 2 supplementary figure 4B. We have included a short section of text in the Results section to describe these additional findings.

With the 5' splice site +4 variation identified, the next step was to determine the underlying molecular mechanisms that dictate the evolution of the various sequence preferences. Some obvious players here are the U1 and U6 snRNAs which directly interact with the 5' splice site during splicing. However, no association was found between these snRNAs and the 5' splice site +4 sequence.The powerful interspecies association mapping was then used to determine whether the presence or absence of METTL16 ortholog or a splicing factor correlated with the 5' splice site +4 sequence variation. Interestingly, a clear association was found between METTL16 and the 5' splice site +4 position; METTL16 presence was associated with +4A at the 5' splice site and METTL16 absence was associated with +4U at the 5' splice site. This is an exciting and significant finding.

We thank the reviewer for these comments on the importance of this study.

Interestingly, the next most significant association with the 5' splice site +4 position was with SNRNP27K. This result makes sense as in the cryo-EM structure of the pre-B spliceosome complex the C-terminal domain of SNRNP27K is found near the region of the U6 snRNA that will interact with the 5' splices site. Absence of SNRNP27K was associated with an increased preference for +4U at the 5' splice site. Now the real power of the interspecies association mapping was demonstrated by investigating whether any association could be determined specifically within the C-terminus of SNRNP27K. Significantly, the methionine 141 position in SNRNP27K was found to be associated with the +4 position of the 5' splice site. This finding fits nicely with previous studies where mutation of M141 caused a shift in 5' splice site selection away from +4A 5' splice sites, to 5' splice sites without +4A. What is not clear is whether M141 is conserved or invariant between all the species that were compared?

M141 is not completely conserved across the species that were compared for the SNRNP27K C-terminus analysis. We did not test positions with very strong sequence conservation, because without variation in both the genotype and phenotype it is not possible to test for an association. We have rephrased the relevant Results and Methods sections to make this point clearer. In addition, we have incorporated a sequence logo to illustrate the degree of conservation of each position in the SNRNP27K C-terminal domain as Figure 5 —figure supplement 1A. Finally, we have included an additional box-plot to illustrate the finding that species which have lost SNRNP27K or have only lost the Methionine equivalent to human SNRNP27K position 141, show a similar preference for +4U at 5’ SSs. This is now included as Figure 5 —figure supplement 1B.

Overall, this result reveals the power of the interspecies association approach and provides interesting and exciting information on the molecular determinants of 5' splice site evolution.

We are grateful to the reviewer for these comments.

The final analysis was to investigate the interaction potentials of the U5 and U6 snRNAs with the 5' splice site in the Sacchromycotina genomes and try to relate this to species with fewer introns and less alternative splicing. Species with low intron numbers and low splicing complexity were revealed to have weaker U5 and U6 anti-correlation potentials and favor +4U at the 5' splice site. On the other hand, species with high intron number and presumably higher splicing complexity featured anti-correlated U5 and U6 snRNA interaction potentials and favored +4A 5' splice sites. This extensive analysis provides novel information on the interactions and splice site properties of species with simple and complex splicing. Again, I see why there is emphasis on the 5' splice site here but a similar analysis with the U2 snRNA and the branch site could also be informative.

We absolutely agree that inter-species association mapping could be applied to other splicing signal phenotypes including 3’ splice sites and intron branchpoints. Accordingly, we raise this subject in the final section of the Discussion. However, branchpoint sequences are challenging to predict with genomic data. Because preliminary analyses suggest independent variation in these other splicing signal phenotypes, we feel a separate focused study is required to properly explain (and substantiate) even the analytical approaches involved. We hope the reviewer would agree that incorporating U2 snRNA and branchpoint variation analyses into this manuscript as well, could detract from the clarity of the conceptual advances that we make here.

Minor commentsShould the Title include SNRNP27K?

We have included SNRNP27K into the revised title.

Should the title specify that it is the evolution of only the 5’ splice site sequence preference being studied here?

Because apostrophes in titles can compromise some scholarly online search engines (https://insights.uksg.org/ar5cles/10.1629/uksg.534), we would prefer not to include 5’ in the title.

Include information on intron number and 5’ splice site interaction potential of U5 and U6 snRNA in the Summary?

We thank the reviewer for this suggestion. We have updated the Summary to include our findings on U5 and U6 interaction potential in species with reduced intron number.

Figure 1C is not referred to in the text?

We apologise for this oversight. We have added references to figure 1C in the appropriate Results section.

Page 8, line 5 – better to say “splicing signal phenotypes”.

We have amended this statement on Page 8 and at other places in the text where related phrasing was made.

What are the other points on Figure 3B? What is the next point below SNRNP27K? Is it U2A’?

The other points on Figure 3B represent Orthofinder orthogroups which contain human orthologs that are known components of the spliceosome. The list of spliceosomal components was taken from Sales-Lee et al. 2021. The third most significant point is indeed the orthogroup containing the human ortholog of U2A’. As we state in the text, however, the correlation of U2A’ with the 5’SS+4 A to U ratio phenotype is no longer significant once METTL16 presence/absence is controlled for, indicating that the correlation of U2A’ with the +4A phenotype is likely explained by similarity in the patterns of gene loss of U2A’ and METTL16.

The second paragraph of the Discussion is vague and lacks a reference. “we could also identify an association with a methionine residue in the conserved C-terminal domain of SNRNP27K orthologs.” There are a few methionines in the C-terminus, which one? Please reference the statement “transcriptome analysis of *C. elegans* SNRP-27 M141T mutants…”

We apologise for the lower quality of writing in this section of the Discussion. We have updated the text, made the statements about the SNRNP27K C-terminus less ambiguous, and added the relevant citations as appropriate.

Reviewer #1 (Significance (Required)):Overall, this is a well written and clearly presented study that provides some key molecular information on the splicing factors involved in the evolution of 5’ splice sites and shows the power of interspecies association studies. Some important conceptual principles have now been defined for the field going forward.

With thank the reviewer for this kind comment on the importance of this work.

The question remains as to whether METTL16 and SNRNP27K are the sole determinants of 5’ splice site preference evolution at +4?

We cannot say for certain that METTL16 and/or SNRNP27K determine the 5’SS +4 phenotype – only that they are correlated with it. In our response to reviewer 3, and in a new Discussion section, we have detailed some of the scenarios that could explain these correlations. We also cannot rule out whether there are changes in the presence/absence (or domain/sequence-level changes) of other, untested proteins that correlate with the 5’SS +4 phenotype and we allude to this in the final section of the Discussion.

One splicing factor that immediately comes to mind is Prp8 where there is extensive evidence for involvement in splice site selection and is clearly in the right location throughout splicing to be involved. This question should at least be discussed but Prp8 would also be a very interesting candidate for the interspecies association mapping.

Prp8 is a core component of spliceosomes and is conserved throughout the Saccharomycotina. For this reason, we were unable to associate splicing phenotypes with Prp8 presence or absence variation at the level of orthogroups. However, we revisited this question posed by the reviewer. Our experience with inter-species association mapping, so far, indicates it works well with orthogroup presence/absence or when straightforward amino acid substitutions can be detected in conserved and hence alignable protein sequence domains. We analysed the conserved U6 snRNA-interacting region of the Prp8 linker domain, which maps close to the 5’ splice site in cryo-EM models, using the profile HMM PF10596 available from Pfam. We found that the majority of this domain was extremely highly conserved with variation in only a few species and positions. The strongest correlation with the +4A to U ratio phenotype was at position 58, which is conserved as a Glycine in all but 8 species (6 Dipodascaceae, 2 CUG-Ser1), that also tend to have a stronger preference for +4A. However, examination of the species contributing to this result (and to similar results at other positions) indicated that in the 6 Dipodascaceae species, this change is part of a larger deletion or replacement that makes the whole linker region align poorly to the model. Hence, the G58 position itself may not be specifically important for the +4 phenotype. Although the wholesale loss or replacement of the U6 snRNA-interacting region in these species is potentially interesting, these larger scale structural changes in a small number of species are difficult to interpret. Therefore, to maintain the focus of the manuscript and the clear links to METTL16 and SNRNP27K that have orthogonal support, we have decided not to add these results to the manuscript but present them here.

**Author response image 1. sa2fig1:** Analysis of the U6-interacting region of the PRPF8 linker domain using pHMMPGLS analysis. (**A**) Stem plot showing the association of individual conserved positions in the U6 snRNA interacting domain of PRPF8 with the 5’SS+4 A to U ratio phenotype in 239 species retaining a detectable PRPF8 ortholog. (**B**) Boxplot showing the distribution of 5’SS+4 A to U ratio phenotypes for species retaining a PRPF8 ortholog with a Glycine at pHMM alignment position 58, compared to those with a different or unalignable residue at this position (X58). Six of the eight species without a Glycine at this position are from the clade Dipodascaceae and have PRPF8 orthologs that align poorly to the pHMM, suggesting that this position itself may not be specifically important for the +4 phenotype.

Also, as mentioned previously, only the 5’ splice site was investigated here and the manuscript could become a more substantial piece of work if the other splice sites were included in some way.

We agree that it will be exciting to apply this approach to other splicing signal phenotypes and in other phylogenetic clades with emerging tree-of-life-scale genomics data. We have included variation in 3’ splice sites in the revised manuscript. As the first of its kind, this study should pioneer a wider use of this approach, by us and others, to understand the mechanisms and functions of molecular interactions not only in splicing but in other areas of biology too.

The obvious audience here are those directly in the splicing field but the overall principles are relevant for evolutionary biologists and those studying organismal complexity.

We thank the reviewer for recognising the broad importance of this work.

My expertise is in yeast and human splicing mechanisms. I do not have the expertise to critically evaluate the bioinformatic pipelines but they were clearly explained and presented.

Reviewer #2 (Evidence, reproducibility and clarity (Required)):In their manuscript, Parker et al. investigate the evolutionary patterns of splice site preference, focusing on the A/U ratio at position A+4 on the 5´ splice site. Building upon prior studies in *S. pombe* and *A. thaliana*, the authors establish a strong correlation between this preference and the co-evolution of the METTL16 U6 snRNA methyltransferase. Furthermore, through inter-species association mapping, they identify the involvement of the splicing factor SNRNP27K in altered A/U ratios and highlight the significance of the residue Met-141 in SNRNP27K for this function. Overall, the paper effectively presents impactful new findings on the evolution of METTL16, U6 snRNA, and splicing.

We thank the reviewer for these kind comments on the importance of our study.

The computational analyses employed in this study are situated outside our field of expertise, preventing us from offering a comprehensive evaluation of the methodology’s appropriateness and rigor. Nonetheless, the identification of METTL16 through the authors’ methods, which aligns with previous research in *S. pombe* and *A. thaliana*, lends support to the validity of their approach. Notably, the close proximity between SNRNP27K and the methylated A43 residue in U6 snRNA within the spliceosome, particularly near Met-141, is an impressive finding. Previous studies have shown that a mutation at position M141T affects splicing at +4A introns, thus providing robust validation for their methods.

We thank the reviewer for these kind comments on our work.

The data presented in this study furnish crucial insights into the role of METTL16, U6 snRNA methylation, and splice site recognition. The authors expand upon recent observations that the “vertebrate conserved region” exists in non-vertebrates, despite the absence of primary sequence homology. These results will serve as a valuable guide for future molecular investigations into U6 snRNA methylation and its mechanisms in splicing. Furthermore, the implications of this paper extend to human evolution, as the plasticity in splicing is an essential factor in the evolution of developmental complexity.

We thank the reviewer for these kind comments.

Minor suggestions for improvement:1. Given the significance of the interaction between U6 snRNA and the intron for understanding the data, it would be beneficial to include a figure illustrating the RNA-RNA base-pairing interactions between U6 snRNA and the 5´ splice site. This addition is particularly important if the paper is intended for publication in a journal with a general readership.

We thank the reviewer for this excellent suggestion. We have included this as Figure 3A.

2. Similarly, the section on U1 snRNA would be more comprehensible with the inclusion of U1 RNA-RNA intron diagrams and improved descriptions of both the figures and the assay. Despite being negative data in the supplement, clarifying this section is essential. As currently written, it is challenging to follow.

We agree that this section is difficult to follow. We have updated the text to improve the readability and included a figure of U1 snRNA:5’SS basepairing as Figure 3 —figure supplement 1A.

3. Whenever possible, consider increasing the figure and font sizes to enhance readability for readers.

We agree that some of the more complex figures can be difficult to read when embedded into a Word document/pdf. We hope that providing high-resolution figures for reading online will mitigate this.

4. In the text, there is no reference to Figure 1C.

We apologise for this oversight. We have resolved this issue with the appropriate references in the Results text.

5. In Figure 5B, the y-axis in the top panel is 8labelled “species,” but the legend only mentions U5/6p as the y-axis. Please revise the legend to include the appropriate information.

We apologise for the confusion caused by our poorly written legend for this plot. We have updated the legend so that the text clearly refers to either the scatter plot or the marginal histograms.

Reviewer #2 (Significance (Required)):The data presented in this study furnish crucial insights into the role of METTL16, U6 snRNA methylation, and splice site recognition. The authors expand upon recent observations that the “vertebrate conserved region” exists in non-vertebrates, despite the absence of primary sequence homology. These results will serve as a valuable guide for future molecular investigations into U6 snRNA methylation and its mechanisms in splicing. Furthermore, the implications of this paper extend to human evolution, as the plasticity in splicing is an essential factor in the evolution of developmental complexity.

We are grateful to the reviewer for these kind comments on the importance of this work.

Reviewer #3 (Evidence, reproducibility and clarity (Required)):In this manuscript, Parker et al. present a nice exploration of the evolutionary and mechanistic relationships between 5ʹ splice site consensus sequences, intron numbers and METTL16/SNRNP27K. By performing inter-species association mapping in Saccharomycotina species, they found that a T in position +4 is strongly associated with the absence of METTL16 (and/or in some cases SNRNP27K or mutations in it). They also provide solid structural modelling data in support of this association.In general, I think this is a very nice manuscript. I only have a few comments, which could be addressed by rewording specific parts and/or improving the current figures.

We are grateful to the reviewer for the kind comments on this work.

1. As the authors acknowledge, a key issue that cannot be fully resolved in this study is causality between the different events investigated. Overall, the authors are careful about this, but there are some exceptions that should be corrected. Probably the most important is in the abstract, where they write: “We conclude that variation in concerted processes of 5’ splice site selection by U6 snRNA is crucial to evolutionary change in splicing complexity”. I suggest they write something more open (and correct), such as: “We conclude that variation in concerted processes of 5’ splice site selection by U6 snRNA is associated with evolutionary changes in splicing complexity”. Similarly, other plausible scenarios should be discussed in the corresponding Discussion section.

We agree with the reviewer that it is not possible to infer the causal relationship between METTL16 absence and 5’SS+4 preference change from the current data. We, therefore, apologise for failing to be more careful in the Summary and Introduction. We have reworded these statements to better reflect what we can currently say about the evolutionary relationship between METTL16 and 5’SS sequence preference.

The correlation between METTL16 absence and 5'SS+4 sequence preference change could most likely be explained by one of several scenarios: (a) sudden loss of METTL16 causes a rapid necessity to change 5'SS sequence preferences. This is unlikely as such rapid change without widespread corresponding 5'SS changes would likely impose a high fitness cost. (b) Changes in 5'SS sequence preference occur first, driven by some other selective pressure, until there is no longer a benefit to retaining the METTL16 gene. (c) Gradual changes in the expression or catalytic efficiency of METTL16 reduce the stoichiometry of U6 snRNA m^6^A modification, which permits gradual change in 5'SS+4 sequence preference until complete loss of the METTL16 no longer imposes a major fitness cost. As we suggest in the Discussion, future work could examine this question by determining whether the METTL16 orthologs found in *Zygosaccharomyces* and *Eremothecium* species, which have altered their 5'SS+4 preference to a U, are expressed and functional. We have updated the Discussion to include a new section that addresses these scenarios.

2. I do not agree with the statement that "The extent of alternative splicing is the best genomic predictor of developmental complexity". To start with, there are many ways to quantify "extent of alternative splicing" and there are also different types of alternative splicing that might have different prevalence and biological impact. Then, this claim is usually related with exon skipping, which is tightly linked with intron length, and that is likely a better prediction of complexity (yet clearly not causative). My concern is: to what extent has this claim been formally and properly assessed by comparing splicing prevalence with other genomic features, such as intergenic region length, intron length, or average distance between enhancer-promoter interactions (arguably the most relevant predictor, in light of many other studies)? Moreover, I found it a bit misleading to frame the work presented in this study as directly related with developmental (or even splicing) complexity. The work is very interesting on its own, and I doubt their findings on +4 position preference in Saccharomycotina has anything to do with developmental complexity (as the Abstract and Introduction seem to imply).

On reflection, we agree with the reviewer. Some of our framing of the text isn’t balanced with other studies on the scaling of alternative splicing with developmental complexity. We have edited the Summary and Introduction sections accordingly and cited other references that broaden the consideration of this subject. We are grateful to the reviewer for this suggestion because the changes we make improve the focus of the manuscript since our findings relate more to splicing simplification than to an understanding of increased developmental complexity.

2. I found Figure 2 and its associated supplementary figure very difficult to follow. I suggest the authors try to improve it and make it clearer. Also, other trees summarizing the results might be helpful.

We apologise for the complexity of these figures. We opted to show phylogenetic trees with phenotypes plotted on the y axis, rather than simply trait histograms or box-plots, because the underlying structure of the tree is important for demonstrating that multiple independent changes in the 5’SS phenotype have occurred in the Saccharomycotina. We have tried to improve the comprehensibility of the figures in the following ways: (a) We have added 5’SS sequence motifs to the x-axis of figure 2B to make what the plot represents clearer, (b) as suggested by the reviewer, we have created a pruned tree showing the 5’SS motifs of a selection of Saccharomycotina species, which demonstrates that the changes in 5’SS+4 position preferences seen in *S. cerevisiae* and *C. albicans* are likely to be a result of convergent evolution. We have added this tree as Figure 2—figure supplement 3.

3. I also found the Results section corresponding to Figure 5B a bit confusing. I would argue (as I think the authors do) that there are two main patterns here: below 500 introns, there is no association, while above 500 introns there is an increasingly negative association (correlation). I think it would help to more explicitly distinguishing these two patterns. Then, for the intron-poor species: is the correlation (or lack of) for species with a T or an A in position +4 different?

We do indeed think that there are two patterns here, as indicated by the reviewer. In the previous version of the manuscript, we separated species into those having an overall preference for A at the +4 position, and those having +4U. By showing regression lines for these two classes, rather than for the general relationship between intron number and U5/6rho, we somewhat imply that the switch in +4 base preference might be causing the loss of correlation between U5/6rho and intron number. However, since essentially all species with a 5'SS +4U preference are intron poor, it seems more likely that these trends are the result of a loss of the negative correlation between intron number and U5/6rho in intron poor species, as suggested by the reviewer. To address this issue, we have replaced the regression lines on Figure 6B with a single loess (locally estimated scatterplot smoothing) regression line for all species and updated the text to make it clearer that we think loss of U5/6rho and +4A preference are separate traits of intron poor species. Although this is not exactly what the reviewer requested, we hope that it satisfies their issue with the analysis.

Reviewer #3 (Significance (Required)):This is a very interesting study that sheds light on an intriguing evolutionary pattern: the change in consensus sequence at position +4 of the 5' splice site. This topic is relevant since it is closely associated with intron loss and splicing efficiency and evolution.

We thank the reviewer for the kind and constructive comments on this study.